# Fasting alleviates metabolic alterations in mice with propionyl-CoA carboxylase deficiency due to *Pcca* mutation
Wentao He[1,8], Hannah Marchuk [1,8], Dwight Koeberl[2], Takhar Kasumov[3], Xiaoxin Chen [4,5,6] & Guo-Fang Zhang [1,7] ✉

Propionic acidemia (PA), resulting from *Pcca or Pccb* gene mutations, impairs propionyl-CoA metabolism and induces metabolic alterations. While speculation exists that fasting might exacerbate metabolic crises in PA patients by accelerating the breakdown of odd-chain fatty acids and amino acids into propionyl-CoA, direct evidence is lacking. Our investigation into the metabolic effects of fasting in $Pcca^{-/-}$(A138T) mice, a PA model, reveals surprising outcomes. Propionylcarnitine, a PA biomarker, decreases during fasting, along with the C3/C2 (propionylcarnitine/acetylcarnitine) ratio, ammonia, and methylcitrate. Although moderate amino acid catabolism to propionyl-CoA occurs with a 23-h fasting, a significant reduction in microbiome-produced propionate and increased fatty acid oxidation mitigate metabolic alterations by decreasing propionyl-CoA synthesis and enhancing acetyl-CoA synthesis. Fasting-induced gluconeogenesis further facilitates propionyl-CoA catabolism without changing propionyl-CoA carboxylase activity. These findings suggest that fasting may alleviate metabolic alterations in $Pcca^{-/-}$(A138T) mice, prompting the need for clinical evaluation of its potential impact on PA patients.

Propionic acidemia (PA) is an inborn error metabolic disorder caused by mutations in the gene responsible for encoding propionyl-CoA carboxylase (PCC). Decreased PCC activity disrupts the conversion of propionyl-CoA into methylmalonyl-CoA and its subsequent entry into the tricarboxylic acid (TCA) cycle, an anaplerotic process within mitochondria. The advancement of PA can give rise to a range of complications, and inadequate management of this condition poses life-threatening risks[1–11].

While the precise pathological mechanisms underlying various complications associated with PA remain incompletely understood, it is clear that the accumulation of propionyl-CoA and its toxic metabolites plays a significant role as pathogenic factors. This is primarily due to the structural similarity between acetyl-CoA (C2 CoA) and propionyl-CoA (C3 CoA), where the increased levels of C3 CoA can result in metabolic crises through competition with C2 CoA. For example, when C3 CoA displaces C2 CoA, it contributes to the formation of propionylglutamate[12], methylcitrate[13–16], and odd-chain fatty acids[17–20], which disrupt normal urea metabolism[21,22], the TCA cycle[14,23], and lipid metabolism[24]. Consequently, the ratio of C3 CoA to

C2 CoA becomes a reliable indicator for assessing the severity of metabolic disturbances in PA patients[25–27].

Propionate, odd-chain fatty acids, side-chain of cholesterol, and propiogenic amino acids such as valine, isoleucine, methionine, and threonine are all metabolic precursors of propionyl-CoA[5,11]. Any modulation of the metabolism of these compounds linked to propionyl-CoA will impact PA. Notably, propionate, primarily produced by the microbiome, and propiogenic amino acids, obtained through dietary sources, are the main contributors to propionyl-CoA production[28]. Consequently, the use of antibiotics to suppress propionate production from the microbiome and the adoption of a protein-restricted diet to reduce propiogenic amino acids are proven effective strategies in alleviating metabolic alterations and improving outcomes in PA disease[29–34].

Extended fasting is typically discouraged for patients with PA due to the belief that fasting may stimulate the oxidation of odd-chain fatty acids and amino acid catabolism to propionyl-CoA, which potentially exacerbates metabolic crises[17,35,36]. However, this recommendation lacks comprehensive

[1]Sarah W. Stedman Nutrition and Metabolism Center and Duke Molecular Physiology Institute, Duke University, Durham, NC 27701, USA. [2]Division of Medical Genetics, Department of Pediatrics, Duke University School of Medicine, Duke University Medical Center, Durham, NC 27710, USA. [3]Northeast Ohio Medical University, Rootstown, OH 44272, USA. [4]Department of Surgery, Surgical Research Lab, Cooper University Hospital and Cooper Medical School of Rowan University, Camden, NJ 08103, USA. [5]Coriell Institute for Medical Research, Camden, NJ 08103, USA. [6]MD Anderson Cancer Center at Cooper, Camden, NJ 08103, USA. [7]Division of Endocrinology, Department of Medicine, Metabolism and Nutrition, Duke University Medical Center, Durham, NC 27701, USA. [8]These authors contributed equally: Wentao He, Hannah Marchuk. ✉e-mail: guofang.zhang@duke.edu

**Fig. 1 | Impact of fasting on metabolism in**
***Pcca$^{-/-}$(A138T) mice. a** Experimental design for
fasting and a 20-min tracing study. **b–e** Changes in
body weight, fat mass, fluid content, and lean mass
observed in both fed and 23-h fasted Pcca$^{-/-}$(A138T)
mice. **f** Plasma levels of 3-hydroxybutyrate (BHB).
**g** Schematic representation of C2 (acetylcarnitine)
and C3 (propionylcarnitine) metabolism from C2-
CoA (acetyl-CoA) and C3-CoA (propionyl-CoA).
CrAT: carnitine acetyltransferase. **h** The ratio of
propionylcarnitine (C3) to acetylcarnitine (C2) in
plasma. **i** Illustration of the competitive interaction
between C2-CoA (acetyl-CoA) and C3-CoA (pro-
pionyl-CoA) in the reaction with oxaloacetate
(OAA) to produce citrate (Cit) and methylcitrate
(Me-Cit). CS: citrate synthetase, TCAC: tri-
carboxylic acid cycle. **j, k** Plasma levels of methyl-
citrate and methylcitrate to citrate ratio. $N = 5$ per
group. The error bar represents the SE. *, **, ***,
and **** denote $p$ values less than 0.05, 0.01, 0.005,
and 0.001, respectively.

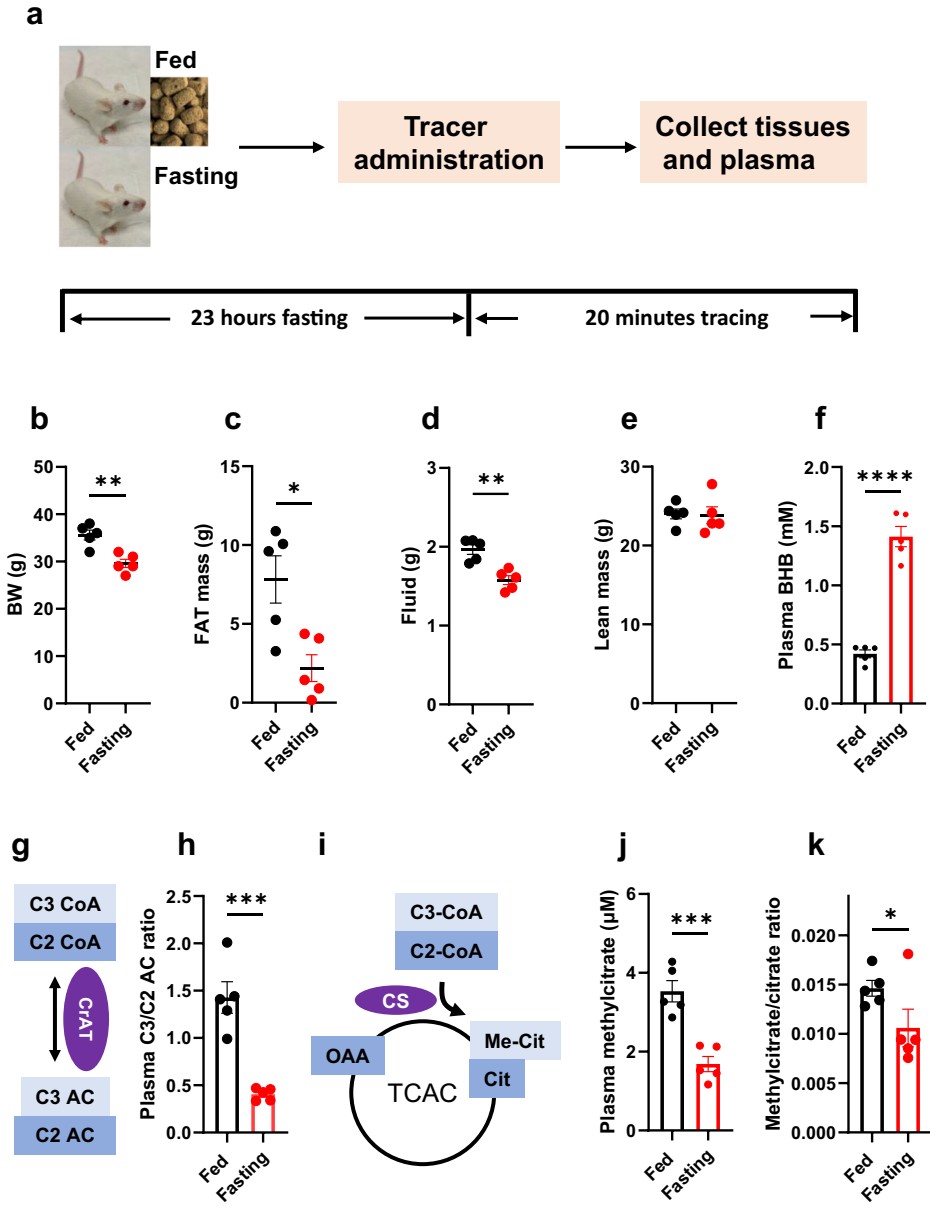

validation in both basic and clinical research[35]. Therefore, there is a pressing
need to address this knowledge gap.

In this study, our objective was to investigate the influence of fasting on
metabolic alterations in a mouse model of PA, specifically Pcca$^{-/-}$(A138T)
mouse. Pcca$^{-/-}$(A138T) mice have ~2% wild-type PCC activity and can
survive to adulthood with the elevations of biomarkers of PA[37]. Even in the
absence of severe symptoms of PA, Pcca$^{-/-}$(A138T) mice remain a valuable
mouse model for studying PA treatments including gene therapy, dual
mRNA therapy, and various drug interventions[26,27,37–39]. Surprisingly, a 23-h
fasting was found to alleviate certain metabolic alterations in Pcca$^{-/-}$(A138T)
mice. The improvement was evident in several metabolic markers, including
reduced levels of propionylcarnitine, ammonia, and methylcitrate, as well as
decreased ratios of propionylcarnitine/acetylcarnitine (C3/C2) and
methylcitrate/citrate. Our data further revealed that the beneficial metabolic
effects of fasting are the result of multiple concurrent events during the
fasting period. Notably, fasting led to a decrease in propionate production
from the microbiome, resulting in reduced propionyl-CoA production.
Simultaneously, fasting induced gluconeogenesis, leading to the utilization
of propionyl-CoA for glucose production. Additionally, fasting promoted
fatty acid oxidation, particularly in the liver, which contributed to the lower

C3/C2 and methylcitrate/citrate ratios. However, the catabolism of amino
acids, such as threonine and valine, to propionyl-CoA remained relatively
stable or showed only moderate changes. All these metabolic shifts occur-
ring during fasting collectively contributed to the improved metabolic
outcomes observed in Pcca$^{-/-}$(A138T) mice.

## Results
### Enhanced fatty acid oxidation and improved PA biomarkers in fasting *Pcca$^{-/-}$* (A138T) mice
Figure 1a outlines the fasting regimen and the design of stable isotope
tracing experiment in Pcca$^{-/-}$(A138T) mice. As anticipated, mice exhibited a
reduction in body weight (6 grams, a 16.9% decrease) after approximately
23 h of fasting, primarily attributed to a decrease in fat mass (5.6 grams, a
72% reduction, Fig. 1b–e). Concurrently, a significant increase in fatty acid
oxidation was observed, as evidenced by a substantial 2.2-fold rise in
3-hydroxybutyrate (BHB) levels in both plasma (Fig. 1f) and all measured
tissues (Supplementary Fig. 1a). Propionyl-CoA competes with acetyl-CoA
in a reaction mediated by carnitine acetyltransferase (CrAT), resulting in the
production of propionylcarnitine (C3 AC) and acetylcarnitine (C2 AC)
(Fig. 1g). These metabolites can enter the circulatory system and be excreted

**Fig. 2 | Attenuated gluconeogenesis drives increased fatty acid oxidation in** *Pcca*[-/-](A138T) **mice. a–d** Body composition of wild-type (WT, N = 10) and *Pcca*[-/-](A138T) mice (N = 6) with comparable body weights (BW). **e** The fold changes of plasma 3-hydroxybutyrate (BHB) levels in *Pcca*[-/-](A138T) mice (N = 11) compared to those in wild-type mice (N = 14) under fed conditions. **f** The fold changes of plasma BHB levels in *Pcca*[-/-](A138T) mice (N = 6) compared to those in wild type mice (N = 5) after 5-h fasting. **g** Glucose production in control and *Pcca*[-/-](A138T) mice following a 5-h fasting period. D2 glucose: [6,6-²H₂] glucose (N = 5). **h** The fold changes in acylcarnitines from C2 to C16 in the organs of *Pcca*[-/-](A138T) mice after a 23-h fasting vs the ones in the organs of fed *Pcca*[-/-](A138T) mice. Quad: quadriceps; WAT: white adipose tissue. N = 5 per group. The error bar represents the SE. *, **, and **** indicate p values less than 0.05, 0.01, and 0.001, respectively.

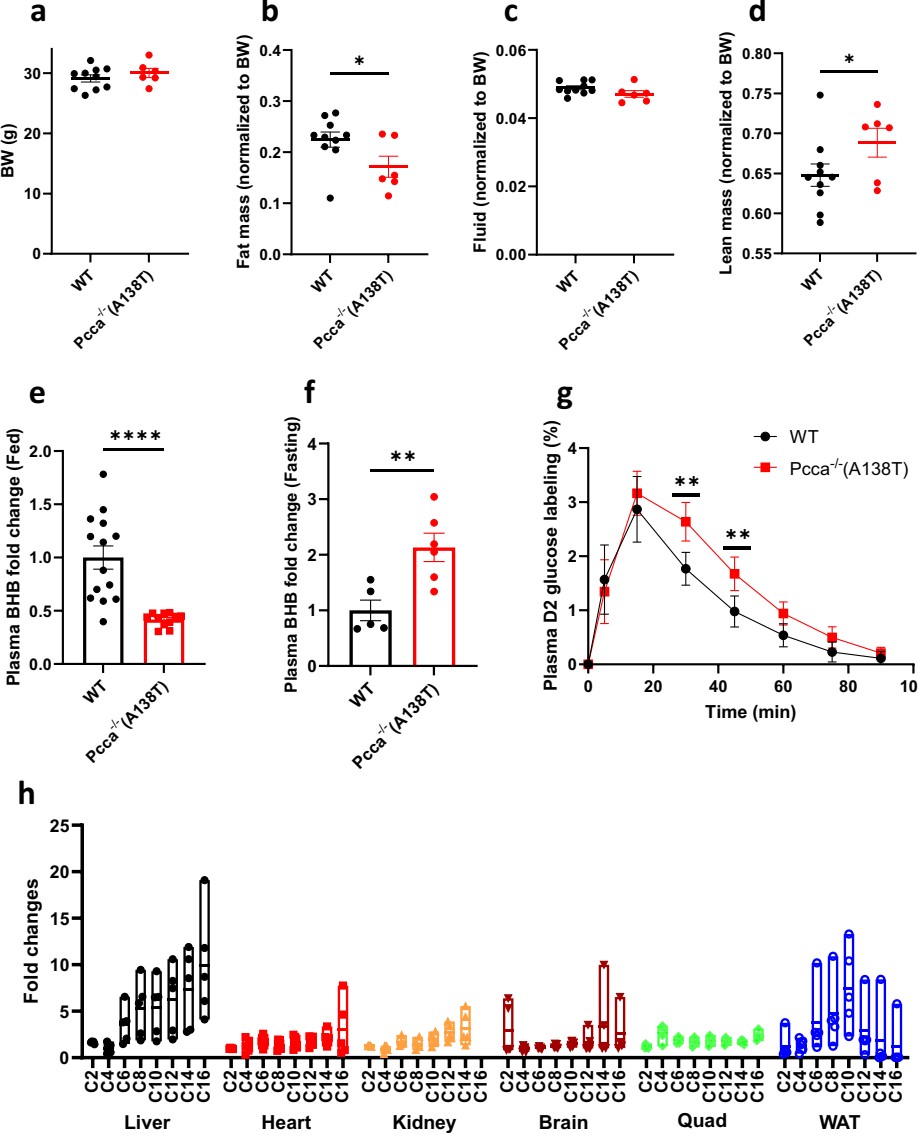

into urine. Consequently, the C3/C2 ratio is often used as an indicator of metabolic disturbances arising from accumulated propionyl-CoA in PA patients. Remarkably, fasting led to a decreased C3/C2 ratio in both plasma (a 62% decrease) (Fig. 1h) and all measured tissues (Supplementary Fig. 1b). The decrease in the C3/C2 ratio in fasted plasma was primarily driven by the substantial decrease in C3 levels (Supplementary Fig. 1c, d). Similar alterations in C3 and C2 levels were also observed in most of the tissues analyzed in this study (Supplementary Fig. 1e, f). Methylcitrate, another reliable biomarker of PA, is formed from oxaloacetate (OAA) by citrate synthetase (CS) through the substitution of acetyl-CoA with propionyl-CoA (Fig. 1i). During fasting, methylcitrate levels decreased by 51%, and the ratio of methylcitrate to citrate also exhibited a significant reduction (27%) in the plasma of fasting *Pcca*[-/-](A138T) mice (Fig. 1j, k). Similar reductions in both methylcitrate levels and the methylcitrate/citrate ratio were predominantly observed in the liver (Supplementary Fig. 1g, h). In addition to the metabolic enhancements mentioned above, fasting notably reduced plasma ammonia levels (Supplementary Fig. 1i).

## The reduced gluconeogenesis in *Pcca*[-/-](A138T) mice promotes fatty acid oxidation in the liver

Interestingly, in the fed condition, *Pcca*[-/-](A138T) mice exhibited a significant reduction (22.7%) in fat mass compared to body weight-matched wild type mice a regular chow diet (Fig. 2a–d). The reduced fat mass in fed

*Pcca*[-/-](A138T) mice is further underscored by notably lower plasma BHB levels. Specifically, plasma BHB levels in *Pcca*[-/-](A138T) mice are approximately 40% of the levels found in their wild type counterparts under fed conditions (Fig. 2e). However, during fasting, it is noteworthy that the increase in plasma BHB levels is considerably higher in *Pcca*[-/-](A138T) mice, amounting to roughly a twofold difference compared to wild type mice (Fig. 2f). This is particularly intriguing as *Pcca*[-/-](A138T) mice maintain lower fat mass (Fig. 2b). This observation may be elucidated by a reduced contribution of propionyl-CoA to gluconeogenesis, as decreased gluconeogenesis results in an increased dependence on fatty acids as the fuel source. As such, we evaluated glucose production during the fasting condition in both wild type and *Pcca*[-/-](A138T) mice using the established stable isotope ([6,6-²H₂]glucose, D2 glucose) approach[40]. Notably, the results clearly indicate that glucose production during fasting is significantly lower in *Pcca*[-/-](A138T) mice when compared to their wild type counterparts (Fig. 2g).

Based on these metabolic changes in PCCA mutated mice, we further quantified changes in acylcarnitine levels within fed and fasted *Pcca*[-/-](A138T) tissues. The alterations in medium- and long-chain acylcarnitine levels serve as additional markers for assessing fatty acid oxidation. Notably, fasting induced a significant rise in acylcarnitines in both the liver and white adipose tissue (WAT), with a particularly pronounced increase in the liver, as, C16 acylcarnitine (AC) levels surged by more than 10-fold (Fig. 2h).

**Fig. 3 | The reduced SCFAs production from the microbiome in fasted *Pcca⁻/⁻*(A138T) mice.**
**a** Plasma collected from the portal vein for SCFAs (short-chain fatty acids) analysis. **b–f** Levels of propionate, acetate, butyrate, pentanoate, and hexanoate in portal vein plasma from both fed and fasted (23-h fasting) *Pcca⁻/⁻*(A138T) mice.
**g** Metabolic alterations in portal vein plasma of fasted (23-h fasting) *Pcca⁻/⁻*(A138T) mice vs fed *Pcca⁻/⁻*(A138T) mice. N = 5 per group. The error bar represents the SE. *, **, and **** indicate *p* values less than 0.01, 0.005, and 0.001, respectively.

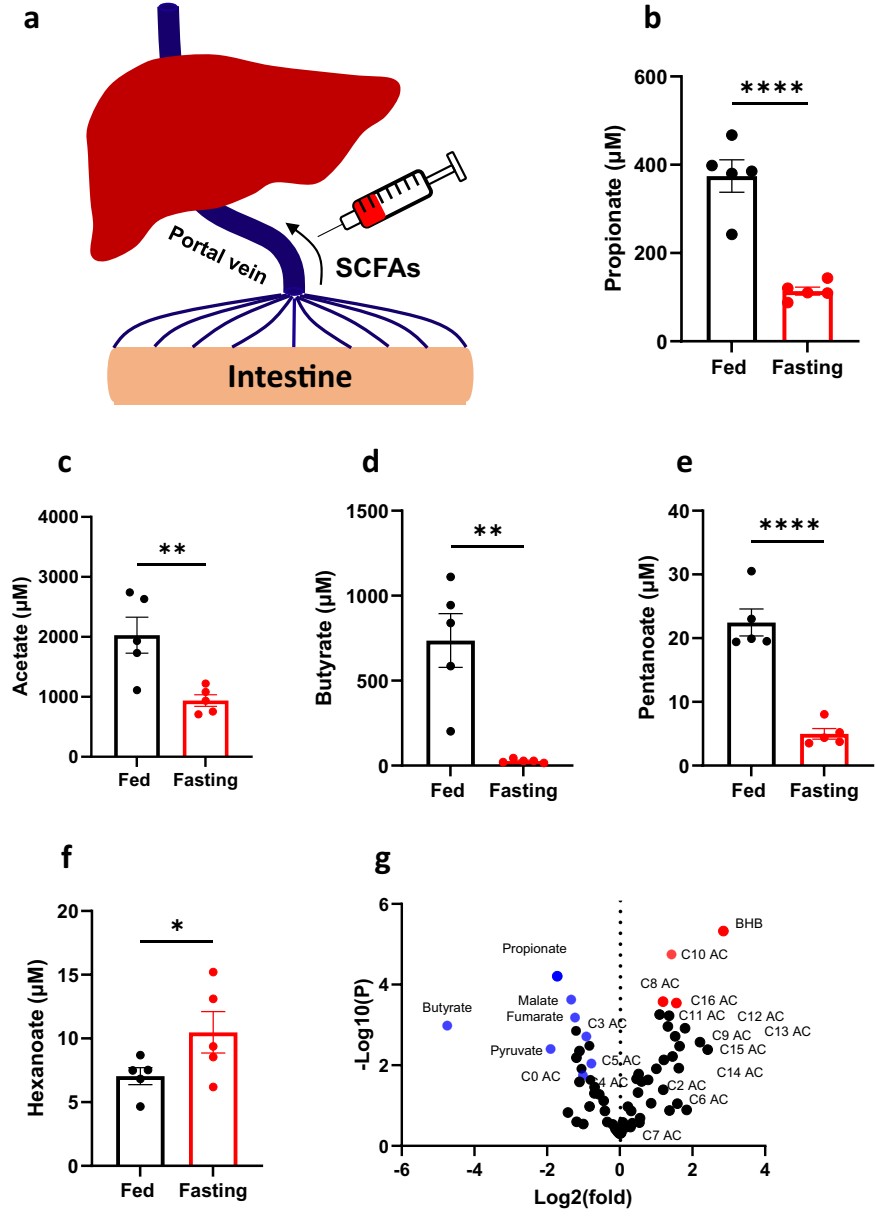

We also observed a notable surge in the oxidation of odd-chain fatty acids across various fasted *Pcca⁻/⁻*(A138T) tissues, as evidenced by the increases in odd-chain acylcarnitines (Supplementary Fig. 2a). Surprisingly, this heightened odd-chain fatty acid oxidation did not result in an increase in propionylcarnitine production (Supplementary Fig. 1c, e). This discrepancy is likely due to the fact that odd-chain fatty acids yield a higher proportion of acetyl-CoA in comparison to propionyl-CoA. For instance, the complete beta oxidation of heptadecanoic acid results in 7 acetyl-CoA molecules and only 1 propionyl-CoA molecule, as depicted in Supplementary Fig. 2b. Consequently, the C3/C2 ratio resulting from the complete oxidation of odd-chain fatty acids (medium- and long-chain length), is still significantly lower than the C3/C2 ratio observed in *Pcca⁻/⁻*(A138T) mice[41].

### Fasting leads to reduced microbiome-derived propionate production

Short-chain fatty acids (SCFAs), which include propionate, are products of gut microbiome. Propionate is primarily metabolized by the liver following absorption in the intestines and transportation through the portal vein. Microbiome-derived propionate is a significant source for propionyl-CoA synthesis[29,42,43]. To assess the changes of the SCFAs ranging from acetate to hexanoate, we measured their levels in the portal vein plasma (as illustrated in Fig. 3a). Interestingly, all the measured SCFAs exhibited a considerable reduction in fasting *Pcca⁻/⁻*(A138T) mice (as depicted in Fig. 3b–e). Propionate levels, for instance, decreased by 75% (as shown in Fig. 3b). The decline in microbiome-derived propionate production induced by fasting, likely contributed to the reduction in propionylcarnitine levels (Supplementary Fig. 1c, e). Hexanoate, a medium-chain fatty acid, primarily originate not from the microbiome[44], and its alteration does not follow the same pattern as short-chain fatty acids (SCFAs) (see Fig. 3f). Conversely, the notable elevation of hexanoate in fasted *Pcca⁻/⁻*(A138T) mice corroborates heightened fatty acid oxidation during fasting, as hexanoate is derived from the hydrolysis of hexanoyl-CoA[45]. The alterations in circulating SCFAs levels, as depicted in Supplementary Fig. 3, particularly acetate (Supplementary Fig. 3b), further substantiate the heightened fatty acid oxidation contributing to SCFAs from the host. Furthermore, we conducted measurements of additional metabolites in the portal vein plasma of both fed and fasting *Pcca⁻/⁻*(A138T) mice to reflect both the contribution of the microbiome produced metabolites as well as circulating levels of these compounds. In the volcano plot, microbiome-derived propionate emerged as the metabolite with the most statistically significant decrease, while BHB

**Fig. 4 | Increased BCAT activity in fasted *Pcca⁻/⁻*(A138T) mice. a** The scheme of $^{15}N$ transfer from [$^{15}N$, $^{13}C_5$]valine to glutamate, leucine, and isoleucine via BCAT (branched-chain amino acid transferase). M1, M5, and M6 denote the presence of 1, 5, and 6 heavy atoms, respectively, within a molecule. **b–d** Labeling of M1 glutamate, M1 isoleucine, and M1 leucine in the plasma of both fed and fasted (23-h fasting) *Pcca⁻/⁻*(A138T) mice. **e** BCAT activity in Quad. $N = 5$ per group. The error bar represents the SE. *, **, ***, and **** indicate $p$ values less than 0.05, 0.01, 0.005, and 0.001, respectively.

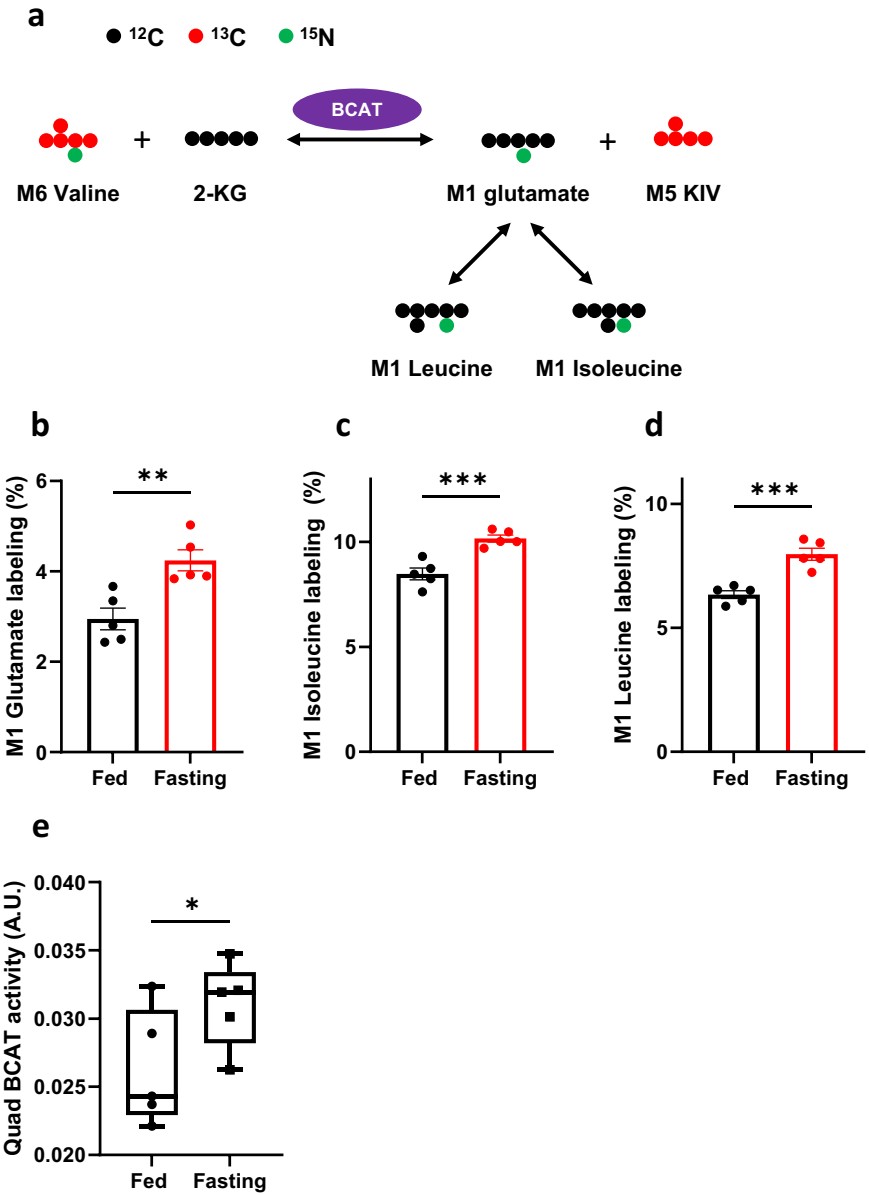

## Catabolism of branched-chain amino acids (BCAAs) to form propionyl-CoA during the fasting state

derived from liver stood out as the most significantly increased metabolite (Fig. 3g). This metabolic profiling outcome reaffirms that fasting has indeed induced a marked reduction in propionate levels and an increase of fatty acid oxidation, particularly in liver. These metabolic alterations collectively contribute to the reductions observed in propionylcarnitine and methylcitrate levels, and the concurrent decrease in the C3/C2 ratio.

The catabolism of BCAAs, namely valine and isoleucine, leads to the generation of propionyl-CoA. Valine's complete catabolism exclusively yields propionyl-CoA, whereas isoleucine catabolism results in the production of both propionyl-CoA and acetyl-CoA. To evaluate the role of BCAA metabolism in propionyl-CoA generation, we employed stable isotope-labeled [$^{15}N$, $^{13}C_5$]valine (M6 valine) to measure the contribution of valine catabolism to propionyl-CoA during fasting conditions. This stable isotope-labeled valine contains both $^{13}C$ and $^{15}N$, enabling us to assess both nitrogen flux (as depicted in Fig. 4a) and carbon flux. Here M0, M1, M2, …, Mn, represents the n number of heavy atoms in a molecule.

Branched-chain amino acid transaminase (BCAT) facilitates the transfer of $^{15}N$ from valine to glutamate, as illustrated in Fig. 4a.

Furthermore, $^{15}N$-labeled glutamate can reciprocally transfer $^{15}N$ back to BCAAs through a reversible reaction mediated by BCAT. We observed a significant increase in M1 labeling ($^{15}N$) of glutamate, isoleucine, and leucine in the plasma of fasted *Pcca⁻/⁻*(A138T) mice (Fig. 4b–d), indicating heightened BCAT-mediated nitrogen flux during fasting. It is worth noting that BCAT expression is high in muscle but lower in the liver[46]. As a result, we evaluated BCAT activity in skeletal muscle, which exhibited a significant increase during fasting, as demonstrated in Fig. 4e. This increase in BCAT activity aligns with the $^{15}N$ labeling data (Fig. 4b–d), suggesting a moderate increase in BCAA transamination in fasted *Pcca⁻/⁻*(A138T) mice.

To gain a comprehensive understanding of BCAA breakdown, we also examined the $^{13}C$ labeling of downstream metabolites of [$^{15}N$,$^{13}C_5$]valine (M6 valine), as portrayed in Fig. 5a. Interestingly, the augmented BCAT activity did not lead to a substantial increase in $^{13}C$ incorporation into downstream metabolites, with the exception of 3-hydroxyisobutyrate (3HIB) (Fig. 5c–e), along with the levels of labeled and unlabeled metabolites in Supplementary Fig. 4a–f. Additionally, the levels of labeled (M3) and unlabeled (M0) propionylcarnitine even exhibited a decrease, as indicated in Supplementary Fig. 4g–h. These metabolic changes in plasma were consistent across various tissues, including the liver, heart, brain, quadriceps (Quad), WAT, and kidney, as demonstrated in Supplementary Fig. 5. In

**Fig. 5 | Valine catabolism to propionyl-CoA remains unaltered during fasting. a** Schematic representation of the metabolism of [$^{15}$N,$^{13}$C$_5$]valine to propionylcarnitine (C3 AC). BCAT: branched-chain amino acid transaminase, BCKDH: branched-chain α-ketoacid dehydrogenase. M3, M4, M5, and M6 denote the presence of 3, 4, 5, and 6 heavy atoms, respectively, within a molecule. **b–e** Measurements of stable isotope labeling of M6 valine, M5 2-ketoisovalerate (KIV), M4 3-hydroxyisobutrate (3HIB), and M3 propionylcarnitine (C3 AC) in the plasma of both fed and fasted (23-h fasting) *Pcca$^{-/-}$*(A138T) mice. *N* = 5 per group. The error bar represents the SE. * indicates *p* values less than 0.05 and 0.001, respectively.

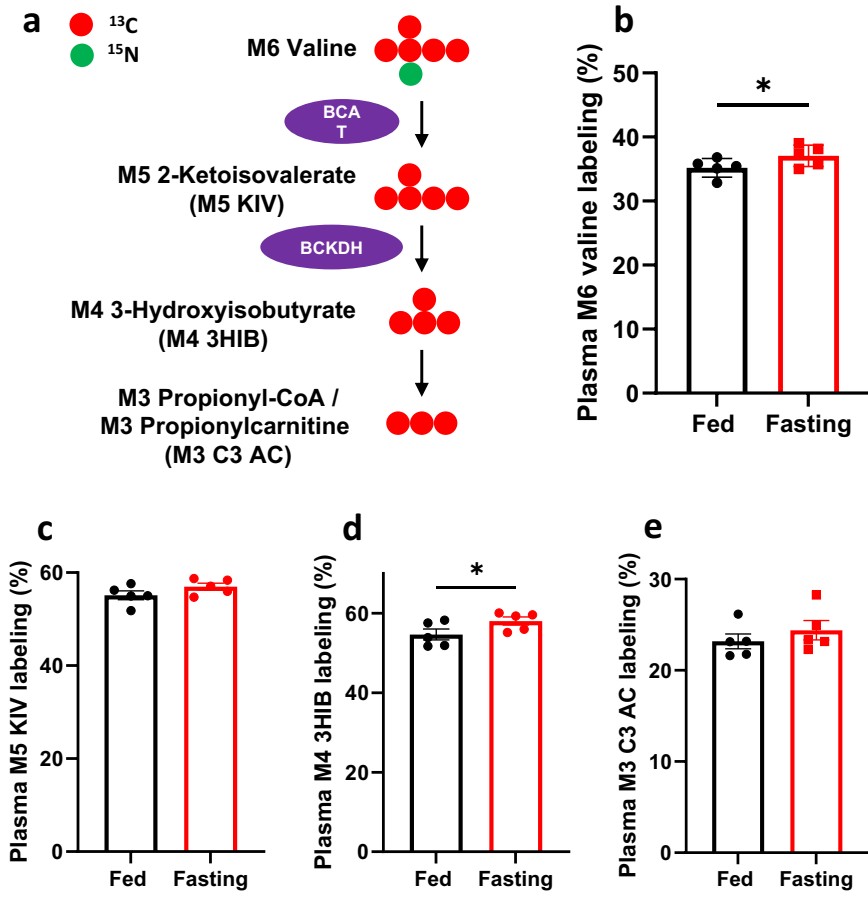

**Fasting increases threonine metabolism in the liver**

In addition to the two BCAAs, it's worth noting that 2-ketobutyrate (2KB), a metabolite originating from threonine and methionine, serves as a metabolic precursor for propionyl-CoA. To investigate potential changes in the metabolism of these two amino acids to 2KB and subsequently to propionyl-CoA during fasting conditions, we conducted experiments employing [$^{13}$C$_4$] threonine (M4 threonine) to trace the metabolic flux of threonine to propionyl-CoA in mice under both fed and fasting states (Fig. 6a). For its direct production of 2KB, labeled threonine was chosen as methionine's catabolism is regulated by several steps to form 2KB as a byproduct. The results are shown in Fig. 6b–f and Supplementary Fig. 6a–d, demonstrating the labeling and concentrations of threonine, 2-hydroxybutyrate (2HB), and propionylcarntine in the plasma of both fed and fasting *Pcca$^{-/-}$*(A138T) mice.

Notably, the labeling of downstream threonine metabolites, 2HB and propionylcarnitine, was significantly higher in the plasma of fasting *Pcca$^{-/-}$*(A138T) mice, as illustrated in Fig. 6c, d. The increased threonine catabolism during fasting was further confirmed by a three-fold elevation in endogenous 2HB (M0 2HB, Supplementary Fig. 6c). Additionally, the production of $^{13}$C-labeled 2HB increased five-fold (Supplementary Fig. 6d), while the endogenous (unlabeled) propionylcarnitine consistently decreased by over 2.5 times (Fig. 6e). The levels of labeled propionylcarnitine in fasting plasma were approximately 60% of those in fed plasma (Fig. 6f). The moderate increase in threonine catabolism did not result in the accumulation of propionyl-CoA/C3 AC, likely due to an elevated utilization of propionyl-CoA, as discussed later in the context of increased gluconeogenesis.

summary, BCAA metabolism exhibited an increased initial step without a significant increase in propionyl-CoA or propionylcarnitine production.

We also measured serine threonine dehydratase (SDS) activity in the liver and skeletal muscle. SDS activity exhibited a dramatic increase in the fasting liver, exceeding 15-fold (Fig. 6g), while it remained unchanged in fasting skeletal muscle (Supplementary Fig. 6e). These findings collectively suggest that the increased threonine catabolism to propionyl-CoA does not lead to an overall increase in propionyl-CoA levels, primarily due to a substantial decrease in the contribution from microbiome-derived propionate and the increase of propionyl-CoA flux to glucose synthesis. The increase in catabolism of threonine is further supported by the labeling and concentration data of threonine, 2HB, and propionylcarnitine in various tissues, including the liver, heart, brain, quad, kidney, and WAT, as depicted in Supplementary Fig. 6f–n. The elevated labeling of 2HB and propionylcarnitine in skeletal muscle can likely be attributed to the release of these metabolites from the liver, where SDS is significantly upregulated (Fig. 6g), suggesting metabolic cross-talk between liver and muscle. Additionally, the upregulated SDS activity in the liver during fasting aligns with previously reported findings in literature[47,48].

**The elevated carbon flux from propionyl-CoA to glucose during fasting condition**

Next, we aimed to evaluate fasting's impact on gluconeogenesis from propionyl-CoA precursors. The incorporation of $^{13}$C into TCA cycle intermediates was undetectable in the $^{13}$C-labeled threonine and valine experiments, likely due to the relatively low level of labeling of propionyl-CoA (less than 30%, as shown in Figs. 5e, 6d and Supplementary Figs. 5j, 6l). This insufficient propionyl-CoA labeling hindered our ability to glean information about the metabolic flux from propionyl-CoA into the TCA cycle and beyond.

To overcome this constraint, we adopted an alternative method by direct labeling of propionyl-CoA with [$^{13}$C$_3$]propionate. For this purpose,

**Fig. 6 | Increased threonine metabolism in fasted *Pcca*⁻/⁻(A138T) mice. a** Schematic illustration of the metabolism of [¹³C₄]threonine (M4 Threonine) to M3 propionylcarnitine (M3 C3 AC). SDS: Serine threonine dehydratase. M0, M3, and M4 denote the presence of 0, 3, and 4 heavy atoms, respectively, within a molecule. **b–d** Isotope labeling of M4 threonine, M4 2-hydroxybutyrate (2HB), and M3 propionylcarnitine (C3 AC), in plasma from both fed and fasted (23-h fasting) *Pcca*⁻/⁻(A138T) mice. **e, f** Levels of M0 C3 AC and M3 C3 AC in plasma from both fed and fasted (23-h fasting) *Pcca*⁻/⁻(A138T) mice. **g** Threonine dehydratase (SDS) activity in the liver of fed and fasted (23-h fasting) *Pcca*⁻/⁻(A138T) mice. *N* = 5 per group. The error bar represents the SE. *, **, ***, and **** indicate *p* values less than 0.05, 0.01, 0.005, and 0.001, respectively.

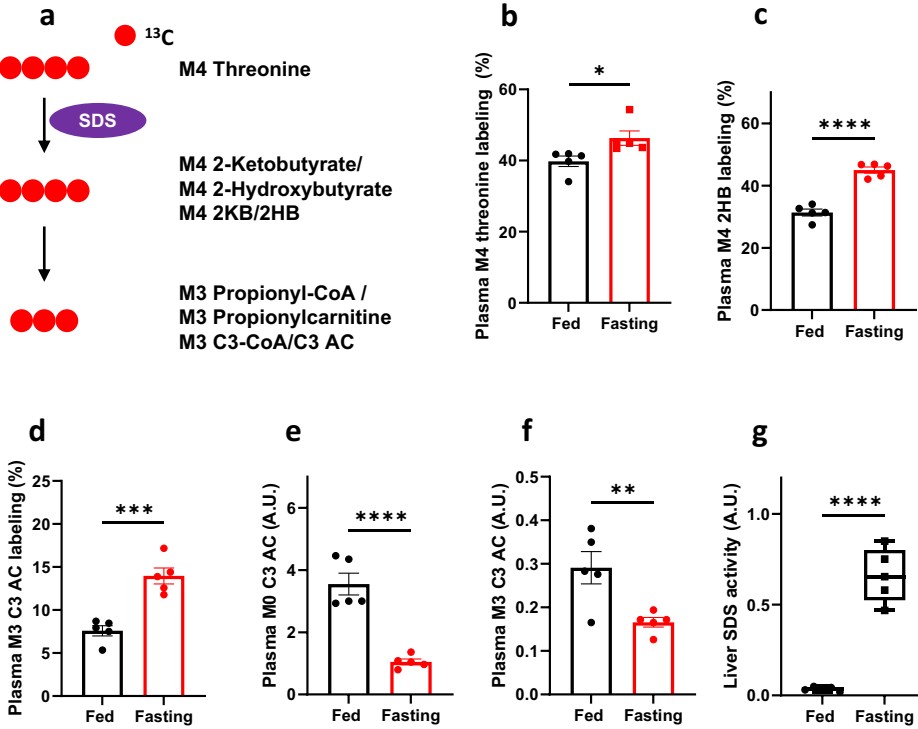

we administered mice with a high dose of [¹³C₃]propionate (500 mg kg⁻¹), as described in our previous report[49]. The ¹³C flow from propionate to propionyl-CoA and its subsequent integration into the TCA cycle and glucose metabolism is depicted in Fig. 7a.

Notably, the plasma propionylcarnitine exhibited substantial labeling from [¹³C₃]propionate (over 80%, Fig. 7b), with even higher labeling in fasting plasma likely due to lower endogenous propionylcarnitine. However, the levels of (un)labeled propionylcarnitine were lower in fasting plasma (Supplementary Fig. 7a, b). The labeling of propionylcarnitine was not significantly different among the tissues (Supplementary Fig. 7c). Despite the high labeling of propionylcarnitine, fasting did not result in increased labeling of TCA cycle intermediates in plasma and tissues (Fig. 7c and Supplementary Fig. 7d–h), except for the brain and WAT. Notably, the changes in TCA cycle intermediate labeling in the brain and WAT were not correlated with PCC activity (Supplementary Fig. 7i, j). The unchanged labeling of TCA cycle intermediates could be attributed to other unlabeled anaplerotic substrates, such as glutamine, glutamate, alanine, lactate, pyruvate, and glycerol, as indicated by their declining levels in plasma (Supplementary Fig. 8a–f)[50,51].

However, the labeling of further downstream metabolites, specifically those involved in gluconeogenesis, such as glucose (over 10-fold increase) and pyruvate (over 1.7-fold increase) derived from the TCA cycle, was significantly higher in fasted plasma (Fig. 7d and Supplementary Fig. 8g). Despite maintaining lower levels of gluconeogenesis compared to wild type counterparts (Fig. 2g), fasting *Pcca*⁻/⁻(A138T) mice experience elevated gluconeogenesis compared to fed controls. The liver, being a major organ responsible for glucose synthesis, was assessed to determine if the increased glucose production affected PCC (propionyl-CoA carboxylase) activity during fasting conditions. Surprisingly, liver PCC activity remained unchanged between fed and fasting conditions (Supplementary Fig. 8h). The unaltered PCC activity following a 23-h fast was similarly observed in the livers of wild-type mice (Supplementary Fig. 8i).

The higher labeled glucose derived from [¹³C₃]propionate in fasting mice suggests an increased carbon flux from propionyl-CoA to glucose production, even in the absence of changes in PCC activity. The liver and kidney, as gluconeogenic organs, showed increased carbon flux from the TCA cycle to glucose, as evidenced by decreased citrate levels (Fig. 7e).

In addition, we assessed the labeling of phosphoenolpyruvate (PEP), pyruvate, lactate, and alanine in various tissues (Supplementary Fig. 8j–m). As expected, the labeling of PEP, pyruvate, and pyruvate's metabolites (lactate and alanine) displayed substantial increases in the liver and moderate increases in the kidney (Supplementary Fig. 8j–m), confirming the enhanced gluconeogenesis.

## Discussion

*PCCA or PCCB* mutations result in impaired propionyl-CoA catabolism and various complications in patients with PA. Therapeutic strategies, such as a protein-restricted diet and antibiotics, aimed at reducing propionyl-CoA synthesis, can improve PA diseases[5]. Fasting is generally discouraged for patients with PA in clinical practice due to the potential acceleration of protein catabolism and odd-chain fatty acid oxidation, which exacerbate the underlying disease through increased propionyl-CoA production[17,35,36]. However, this assertion lacks supporting evidence from research. This work aimed to investigate fasting-induced metabolic changes to gain a deeper understanding of the pathological mechanisms in PA.

In this study, a 23-h fasting regimen was implemented to simulate practical scenarios that might occur in humans. Notably, fasting resulted in several significant metabolic changes, including a decrease in fat mass, an increase in ketone production, and alterations in acylcarnitine levels, indicating a shift toward fatty acid utilization. The most remarkable finding was the improvement in biomarkers of PA, such as reduced propionylcarnitine, methylcitrate (methylcitrate/citrate ratio), and plasma ammonia, along with a decline in the C3/C2 ratio. These metabolic changes during fasting suggested an improvement in the metabolic alterations occurring in PA. It is established that protein-restriction diet alleviates metabolic changes in patients with PA[5]. However, it's important to highlight that fasting encompasses more than just protein restriction. While a protein-restricted diet reduces protein and amino acid intake to mitigate propionyl-CoA synthesis from propiogenic amino acids, fasting triggers broader metabolic alterations. These include enhanced fatty acid oxidation, gluconeogenesis, inhibited propionate production from the microbiome, and increased protein breakdown due to the absence of food intake.

The decrease in methylcitrate and the methylcitrate/citrate ratio can be attributed to the rebalancing of acetyl-CoA and propionyl-CoA levels

**Fig. 7 | Enhanced gluconeogenesis increases the disposal of propionyl-CoA in fasted *Pcca*<sup>-/-</sup>(A138T) mice. a** Schematic representation of the catabolism of [$^{13}C_3$]propionate to the tri-carboxylic acid (TCA) cycle and gluconeogenesis. OAA: oxaloacetate, PEP: phosphoenolpyruvate. **b–d** Stable isotope labeling of M3 propionylcarnitine (M3 C3 AC), average carbon labeling of citrate (Citrate avg C labeling), and average carbon labeling of glucose (Glucose avg C labeling) from [$^{13}C_3$] propionate in the plasma of both fed and fasted (23-h fasting) *Pcca*<sup>-/-</sup>(A138T) mice. **e** Changes in citrate levels in the liver, heart, brain, Quad, WAT, and kidney after a 23-h fasting. M3 denotes the presence of 3 heavy atoms within a molecule. Quad: quadriceps; WAT: white adipose tissue. N = 5 per group. The error bar represents the SE. * and *** indicate *p* values less than 0.05, 0.01, 0.005, and 0.001, respectively.

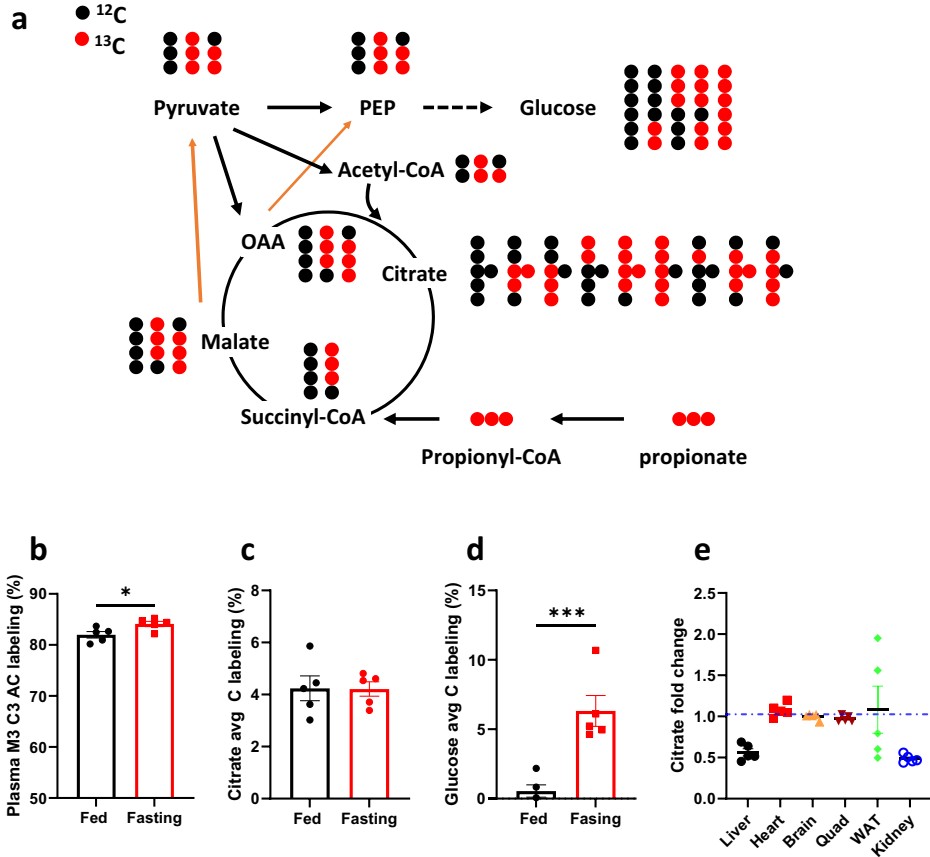

during fasting. In normal conditions, acetyl-CoA is approximately ten times more abundant than propionyl-CoA[41,52]. However, in PA, propionyl-CoA levels approach those of acetyl-CoA[26,27,41]. Due to their structural similarity, propionyl-CoA can replace acetyl-CoA in metabolic reactions, such as the formation of propionylglutamate, odd-chain fatty acids, and methylcitrate. Consequently, the decrease in methylcitrate and the methylcitrate/citrate ratio indicates a restoration of the acetyl-CoA and propionyl-CoA balance during fasting.

The decrease in the C3/C2 ratio during fasting may result from the increased synthesis of acetyl-CoA and a decrease in propionyl-CoA, or a combination of both factors. Enhanced fatty acid oxidation during fasting likely contributes to increased acetyl-CoA production. The impaired gluconeogenesis in *PCCA*-mutated mice drove higher fatty acid oxidation, which is evidenced by high ketones as shown in Fig. 1f and Supplementary Fig. 1a. This is also consistent with clinical observation that patients with PA also produce significantly higher levels of ketones during fasting compared to healthy individuals[53]. Fasting-induced fatty acid oxidation not only enhances the oxidation of even-chain fatty acids but also odd-chain fatty acids, potentially leading to increased propionate production, as suggested by ref. 19. However, their study exclusively measured propionate production and did not assess broader metabolic changes, such as the levels of acetyl-CoA/acetylcarnitine, propionyl-CoA/propionylcarnitine, and methylcitrate during fasting. It's worth noting that propionyl-CoA is generated from the complete beta-oxidation of an odd-chain fatty acid, resulting in the production of multiple acetyl-CoA molecules (7 acetyl-CoA in C17 fatty acid) per odd-chain fatty acid before the formation of a single molecule of propionyl-CoA (Supplementary Fig. 2b). The elevated acetyl-CoA/propionyl-CoA ratio resulting from both even-chain and odd-chain fatty acid oxidation may have beneficial effects in alleviating metabolic alterations in PA.

Additionally, it's important to highlight that Sbai's study employed metronidazole to treat patients prior to fasting[19], which could have minimized the impact of fasting on propionate production from the microbiome. Surprisingly, we found that propionate was significantly reduced in portal vein during fasting, indicating a decrease in the production of SCFAs, including propionate, by the microbiome[54]. This is likely a result of decreased food presence in the intestine and shifts in microbiome composition. This reduction in propionate partially contributed to the observed decrease in propionylcarnitine levels in *Pcca*<sup>-/-</sup>(A138T) mice during fasting. The distinct distributions of SCFAs in the portal vein (Fig. 3b–e) and circulation (Supplementary Fig. 3b–d) confirm the reduction in SCFAs production by the microbiome and an increase in fatty acid oxidation in fasted *Pcca*<sup>-/-</sup>(A138T) mice.

While changes in propionyl-CoA carboxylase (PCC) activity were not detected during fasting, the study revealed an enhanced gluconeogenesis process that resulted in an increased carbon flux from propionyl-CoA to glucose. This was evident from the [$^{13}C_3$]propionate tracing data (Fig. 7d). The heightened propionyl-CoA flux to glucose during fasting played a substantial role in its disposal. According to Leonard's research, approximately 25% of propionyl-CoA production is attributed to propionate derived from the microbiome[55]. The reduction in propionate levels during fasting accounted for an approximate 17% decrease in propionyl-CoA production. Consequently, the upsurge in gluconeogenesis from propionyl-CoA could lead to a significant reduction (~47% or less) in propionyl-CoA in the fasted *Pcca*<sup>-/-</sup>(A138T) mice.

Furthermore, the study explored the catabolic contribution to propionyl-CoA from BCAAs, which initiates within muscle due to its high BCAT activity compared to the liver's low BCAT activity. Subsequently, the catabolic intermediates of BCAAs are transported to the liver for complete catabolism[46]. Stable isotope tracing analysis of valine metabolism suggested the unchanged propionylcarnitine production from BCAAs, although BCAT in skeletal muscles was up regulated during fasting.

Threonine and methionine are additional sources of propionyl-CoA through the pathway involving 2KB. In our study, we employed [$^{13}C_4$]

threonine to examine the metabolism of threonine into propionyl-CoA during fasting. Notably, the liver, a primary organ for threonine's initial catabolism, demonstrated a notable increase in SDS activity, facilitating threonine's conversion into propionyl-CoA across all detected organs during fasting, likely due to the inter-organ cross-talk mediated by downstream metabolites, although muscle SDS activity remained unchanged. However, despite this heightened threonine metabolism, the overall propionyl-CoA/C3 AC levels did not exhibit a significant increase. This result can be attributed to the reduced contribution from other sources, such as propionate produced by the microbiome, and the increased utilization of propionyl-CoA for gluconeogenesis.

A 23-h fasting resulted in a noticeable improvement of metabolism of propionyl-CoA in $Pcca^{-/-}$(A138T) mice. However, it's important to acknowledge the limitations of the study. These include the omission of a comprehensive exploration of potential risks associated with fasting in the context of PA. Additionally, the Pcca$^{-/-}$(A138T) mouse model, representing one of the gene mutations observed in human patients with a milder phenotype[38,49,56,57], may not fully recapitulate the diverse conditions (various genotypes, disease severity, and metabolic crises) encountered in human PA patients. The difference between fasting in this work and protein restriction diet in clinical practice warrants further investigation.

In summary, fasting in mice with *PCCA* mutation led to a substantial reduction in propionate levels in the portal vein and increased metabolic flux from propionyl-CoA to gluconeogenesis. These changes decreased the overall circulating propionylcarnitine. Simultaneously, the significant increase in fatty acid oxidation elevated acetyl-CoA levels, rebalanced the C2 and C3 ratio, and suppressed methylcitrate synthesis and ammonia levels. These findings challenge traditional recommendations for PA patients and suggest the potential metabolic benefits of fasting. However, it's essential to highlight that the applicability of these findings from a non-severe mouse model to human patients necessitates further investigation.

## Methods

### Reagents and chemicals

[$^{15}$N, $^{13}$C$_5$]valine, [$^{15}$N, $^{13}$C$_4$]threonine, [$^{13}$C$_3$]propionate, D9 Carnitine, [2,2,2-$^2$H$_3$-1,2-$^{13}$C$_2$]acetate, 20 μM [2,2,3,3,3-$^2$H$_5$]propionate, [2,2,3,3,4,4,4-$^2$H$_7$]butyrate, [2,2,3,3,4,4,5,5,5-$^2$H$_9$]pentanoate, [6,6'-$^2$H$_2$]glucose (D2 glucose), and [2,2,3,3,4,4,5,5,6,6-$^2$H$_{11}$]hexanoate were from Cambridge Isotope Laboratories (Tewksbury, MA). All other chemicals were from Sigma (St. Louis, MO).

### Mice tracing experiment with intraperitoneal injections

All animal protocols obtained approval from Duke University's IACUC Committee. We have complied with all relevant ethical regulations for animal use. $Pcca^{-/-}$(A138T) mice (*FVB* strain, male, 16–20 weeks old) were divided into two groups: a control group (fed, $n = 5$) and a fasting group (fasting, $n = 5$), with the fasting group undergoing a 23–24 h fasting period prior to the tracing experiment. Following this, all mice received a single bolus injection intraperitoneally, consisting of one of the following tracers: [$^{15}$N, $^{13}$C$_5$]valine (100 mg kg$^{-1}$), [$^{15}$N, $^{13}$C$_4$]threonine (100 mg kg$^{-1}$), or [$^{13}$C$_3$]propionate (500 mg kg$^{-1}$). This injection was administered 20 min before the mice were sacrificed for plasma and organ collection. Anesthesia was induced using 5% isoflurane, and blood samples were drawn from both the inferior vena cava and the portal vein. The collected blood samples were then centrifuged for 5 min at $12,000 \times g$ to obtain plasma. Simultaneously, tissues were rapidly excised, snap-frozen in liquid nitrogen, and subsequently pulverized. All samples were stored at $-80\,°C$ until further analysis.

### GC-MS for metabolite profile

We profiled the metabolic changes in organs and plasma using our previously published GC-MS method[49,58,59]. Briefly, approximately 20 mg of tissue was spiked with 0.2 nmol of norvaline and 0.4 nmol [$^2$H$_9$]L-carnitine or mixed stable isotope labeled metabolites as internal standards and then subjected to extraction through the standard Folch method with 400 μl methanol, 400 μl H$_2$O, and 400 μl chloroform. A 20 μl plasma was

spiked with the corresponding internal standard. Then 500 μl methanol was added and vortex, followed by 500 μl acetonitrile and vortex. Centrifuged for 20 min. The upper phase, approximately 300 μl in volume, was transferred to a fresh Eppendorf vial and subsequently evaporated using nitrogen gas. The resulting dried residues underwent sequential derivatization with methoxyamine hydrochloride and N-tert-butyldimethylsilyl-N-methyltrifluoroacetamide (TBDMS). Specifically, 40 μl of methoxyamine hydrochloride (2% (w/v) in pyridine) was added to the dried residues, followed by incubation for 90 min at 40 °C. Subsequently, 20 μl of TBDMS with 1% tert-butylchlorodimethylsilane was added, and the mixture was incubated for an additional 30 min at 80 °C. The derivatized samples were then centrifuged for 10 min at $12,000 \times g$, and the supernatants were transferred to GC vials for further analysis. For GC/MS analysis, we employed an Agilent 7890B GC system with an Agilent 5977 A Mass Spectrometer, following the methodology described in our previous work[49,58,59]. Specifically, 1 μl of the derivatized sample was injected into the GC column. The GC temperature gradient began at 80 °C for 2 min, increased at a rate of 7 °C per minute to 280 °C, and was maintained at 280 °C until the 40-min run time was completed. The ionization was conducted via electron impact (EI) at 70 eV, with Helium flow at 1 ml min$^{-1}$. Temperatures of the source, the MS quadrupole, the interface, and the inlet were maintained at 230 °C, 150 °C, 280 °C, and 250 °C, respectively. Mass spectra (m/z) in the range of 50 to 700 were recorded in mass scan mode.

### LC-MS/MS for acylcarnitine profile

Tissue acylcarnitines were methylated and profiled using a modified LC-MS/MS method[24,49,58]. The tissue sample extracts (300 μl) from the previous sample preparation were completely dried using nitrogen gas. The dried residues were then methylated with a 3 M HCl methanol solution (100 μl) at 50 °C for 25 min. After methylation, the samples were once again dried completely using nitrogen gas and then reconstituted in 20 μl of methanol and 60 μl of water. The derivatized samples were subsequently analyzed using an LC-QTRAP 6500$^+$-MS/MS (Sciex, Concord, Ontario). A gradient HPLC method with two mobile phases (mobile phase A was 98% water with 2% acetonitrile and 0.1% formic acid and mobile phase B was 98% acetonitrile with 2% H$_2$O and 0.1% formic acid) was adopted to run with an Agilent Pursuit XRs 5 C18 column ($150 \times 2.0$ mm). The gradient started with 0% B within the first 2 min and then increased to 80% at 13 min. The column was washed out by 90% B for 4 min and equilibrated with initial condition (2% B) for 5 min before next injection. The flow rate was 0.4 ml minute$^{-1}$ and the column oven was set at room temperature. The injection volume was 2 μl. The parameters for Sciex QTRAP 6500+ mass spectrometry were optimized as follows: DP: 33 V, EP 10 V, CXP: 10 V, source temperature: 680 °C, gas 1: 65, gas 2: 65, curtain gas: 35, CAD: 10, and ion spray voltage: 5500 V. The Q1 of all the methylated acylcarnitines was scanned from m/z 218 to m/z 444 with the same fragment (Q3) at m/z 99. L-carnitine had the ion transition of Q1 (m/z 176) and Q3 (m/z 85 or m/z 117). [$^2$H$_9$]L-carnitine has the shifted Q1 at m/z 179 or m/z 185 with the same Q3 at m/z 85 or m/z 117.

### Tissue PCC activity assay

The approach to measure PCC activity was adopted from published work[60]. Twenty milligrams of tissue were homogenized in 1 ml of a 50 mM potassium phosphate buffer (pH 7.4). Ten microliters of tissue extract were used, and 100 μl of an enzyme reaction mixture was added. This mixture consisted of 100 mM Tris-HCl (pH 7.5), 5 mM MgCl$_2$, 1 mM DTT, 10 mM KCl, 40 mM NaHCO$_3$, 1 mM Biotin, and 6 mM ATP. The reaction was initiated by adding 10 μl of 11.8 mM propionyl-CoA and carried out at 37 °C for 30 min. Subsequently, 10 μl of the reaction mixture was combined with 10 μl of 0.01 mM D9 pentanoyl-CoA as an internal standard (IS), followed by mixing with 50 μl of 200 mM formic acid to terminate the reaction. The mixture was then centrifuged at $13,000 \times$ rpm for 10 min. The supernatants were analyzed using LC-QTRAP 6500$^+$-MS/MS (Sciex, Concord, Ontario)[52,61]. Ion chromatograms of propionyl-CoA, methylmalonyl-CoA,

and D9 pentanoyl-CoA were extracted and quantified based on m/z values of 824/317, 868/361, and 861/354, respectively.

## L-serine threonine dehydratase (SDS) activity assay

The SDS activity assay was modified from published work[62]. A 20 mg tissue sample was homogenized in 1 ml of a 50 mM potassium phosphate buffer (pH 7.4). In a 2 ml vial, 100 µl of a 50 mM Tris buffer (pH 8.5) was combined with 30 µl of 100 mM KCl, 60 µl of a 5 mg ml⁻¹ solution of PLP (pyridoxal 5'-phosphate hydrate), and 100 µl of the tissue homogenate. The reaction was initiated by adding 30 µl of 100 mM threonine, and incubated at 37 °C for 2 h. After incubation, 20 µl of the reaction mixture was extracted, and 200 µl of acetonitrile along with 40 µl of a 0.01 mM M5 KIV (as an internal standard) were added. The solution was then centrifuged for 15 min, and the supernatants were collected and dried completely. The dried residues containing unreacted threonine and reaction product 2KB were subsequently derivatized with methoxyamine hydrochloride and N-tert-butyl-dimethylsilyl-N-methyltrifluoroacetamide (TBDMS) sequentially. Specifically, 40 µl of a 2% (w/v) methoxyamine hydrochloride in pyridine was added to the dried residues and incubated for 90 min at 40 °C. This was followed by the addition of 20 µl of TBDMS with 1% tert-butylchlorodimethylsilane and incubation for 30 min at 80 °C. The samples were then centrifuged for 10 min at $12{,}000 \times g$, and the supernatants of the derivatized samples were transferred to GC vials for further analysis. GC/MS analysis was conducted using an Agilent 7890B GC system with an Agilent 5977 A Mass Spectrometer, following previously established procedures. Specifically, 1 µl of the derivatized sample was injected into the GC column. The GC temperature gradient initiated at 80 °C for 2 min, increased to 280 °C at a rate of 7 °C per minute, and was maintained at 280 °C until the completion of a 40-min run. Electron impact (EI) ionization at 70 eV with a helium flow of 1 ml min⁻¹ was used. The source, MS quadrupole, interface, and inlet temperatures were maintained at 230 °C, 150 °C, 280 °C, and 250 °C, respectively. Mass spectra (m/z) ranging from 50 to 700 were recorded in mass scan mode. Threonine, 2KB, and M5 KIV were quantified based on the peak areas of ions at m/z of 404, 188, and 207, respectively.

## BCAT activity assay

A 20 mg tissue sample was homogenized in 1 ml of a 50 mM potassium phosphate buffer (pH 7.4). In a 2 ml vial, 100 µl of the same potassium phosphate buffer (pH 7.4) was added. To this, 60 µl of a 50 mM 2KG solution and 100 µl of the tissue homogenate were included. The reaction was initiated by adding 60 µl of a 50 mM valine solution and incubated at 37 °C for 30 min. Following incubation, 20 µl of the reaction mixture was extracted, and 200 µl of acetonitrile, along with 40 µl of a 0.01 mM M5 KIV solution as an internal standard (IS), were added. The solution was then centrifuged for 15 min, and the supernatants were collected and completely dried. Subsequently, the dried residues were derivatized with methoxyamine hydrochloride and N-tert-butyldimethylsilyl-N-methyltrifluoroacetamide (TBDMS) sequentially. Specifically, 40 µl of a 2% (w/v) methoxyamine hydrochloride solution in pyridine was added to the dried residues and incubated for 90 min at 40 °C. This was followed by the addition of 20 µl of TBDMS with 1% tert-butylchlorodimethylsilane and incubation for 30 min at 80 °C. The samples were then centrifuged for 10 min at $12{,}000 \times g$, and the supernatants of the derivatized samples were transferred to GC vials for further analysis. GC/MS analysis was conducted as previously described using an Agilent 7890B GC system with an Agilent 5977 A Mass Spectrometer. Specifically, 1 µl of the derivatized sample was injected into the GC column. The GC temperature gradient began at 80 °C for 2 min, increased to 280 °C at a rate of 7 °C per minute, and was maintained at 280 °C until the completion of a 40-min run. Ionization was achieved by electron impact (EI) at 70 eV with a helium flow of 1 ml min⁻¹. Temperatures of the source, the MS quadrupole, the interface, and the inlet were maintained at 230 °C, 150 °C, 280 °C, and 250 °C, respectively. Mass spectra (m/z) ranging from 50 to 700 were recorded in mass scan mode. KIV and M5 KIV were quantified based on the peak areas of ions at m/z of 202 and 207, respectively.

## Body composition measurement

Body compositions for both fed and fasting $Pcca^{-/-}$(A138T) mice (16–20 weeks old) were determined using a Bruker NMR Body Composition – LF90 Instrument. Fat mass, lean mass, and fluid were normalized to body weight.

## Glucose production in wild-type and $Pcca^{-/-}$ (A138T) mice

The approach for measuring glucose production utilized stable isotopes and was adapted from the research conducted by Previs[40]. A total of 10 mice (16–20 weeks old), comprising both wild type (n = 5) and $Pcca^{-/-}$(A138T) mice (n = 5), were subjected to a 5-h fasting period prior to commencing glucose production measurements. To trace glucose production, an intra-peritoneal bolus of $[6,6^{-2}H_2]$glucose (D2 glucose, 0.04 g kg⁻¹) was administered, and blood samples of approximately 5 µl were collected at multiple time intervals: 5, 15, 30, 45, 60, 75, and 90 min. The kinetics of D2 glucose labeling changes in the blood were indicative of glucose production in the mice. The D2 glucose labeling in plasma was quantified using LC-QExactive⁺- Orbitrap-MS[49]. In this process, 2 µl of plasma was placed in an Eppendorf tube and subjected to Folch extraction, using the following solvents: 200 µl of methanol, 200 µl of distilled $H_2O$, and 200 µl of chloroform. The sample mixture was vortexed and then centrifuged for 20 min at $14{,}000 \times g$. The upper phase, comprising approximately 350 µl, was completely dried under nitrogen gas at 37 °C. The resulting dried residue was reconstituted with 60 µl of distilled water, vortexed, and transferred to an LC vial for subsequent LC-MS analysis. For the glucose labeling assay, an LC-Q-Exactive⁺-Orbitrap-MS instrument was employed. The Vanquish Binary Pump was used to deliver a mobile phase consisting of 98% $H_2O$ and 2% methanol with 0.01% formic acid, at a flow rate of 0.5 ml min⁻¹ in an isocratic elution mode. The column utilized was a Microsorb-MV C18 column (100 × 4.6 mm, 3 µm) with a C18 guard column, maintained at 40 °C in a column oven compartment. The autosampler was kept at 5 °C, and the injection volume was 1 µl. The entire analytical run lasted 10 min.

The Q-Exactive⁺- Orbitrap-MS, equipped with a HESI probe, was set up with the following parameters: a heat temperature of 425 °C, sheath gas at 30, auxiliary gas at 13, sweep gas at 3, a spray voltage of 3.5 kV in positive mode, a capillary temperature of 320 °C, and an S-lens setting of 45. A full m/z scan range was configured from 60 to 900, with a resolution set at 70,000 at m/z 200. The maximum injection time (max IT) was set at 200 ms, and the automated gain control (AGC) was targeted at $3 \times 10^6$ ions. The sodium adduct of glucose was employed to assess M0, M1, M2, and D2 glucose at m/z 203.0530, 204.0563, 205.0573, and 205.0653, respectively. This method was also employed for glucose labeling assay in the $[^{13}C_3]$propionate tracing experiment.

## Short-chain fatty acid assay

The LC-MS/MS method was adapted to analyze short-chain fatty acids, including propionate, in plasma[49]. A 30 µl plasma sample was combined with 30 µl of internal standards, comprising 200 µM $[2,2,2^{-2}H_3-1,2^{-13}C_2]$ acetate (M5 acetate), 20 µM $[2,2,3,3,3^{-2}H_5]$propionate (D5 propionate), 20 µM $[2,2,3,3,4,4,4^{-2}H_7]$butyrate (D7 butyrate), 20 µM $[2,2,3,3,4,4,5,5,5^{-2}H_9]$pentanoate (D9 pentanoate), and 20 µM $[2,2,3,3,4,4,5,5,6,6,^{-2}H_{11}]$hexanoate (D11 hexanoate). Acetonitrile (1 ml) was added to precipitate proteins. After vortexing and centrifuging the samples at $12{,}000 \times g$ for 20 min, the supernatant was transferred to a new Eppendorf vial and dried completely under nitrogen gas. The dried residue was then resuspended in 50 µl of HPLC water and 20 µl each of 3-Nitrophenylhydrazine hydrochloride (EDC, 120 mM) and N-(3-Dimethylaminopropyl)-N′-ethylcarbodiimide (3-NPH, 200 mM) for derivatization at 40 °C for 30 min. After centrifugation for 10 min at $1200 \times g$, the supernatant was transferred to an LC-MS/MS vial for analysis. The LC-MS/MS analysis was performed using a Sciex QTRAP 6500⁺- MS connected to a Sciex AD UHPLC. Separation was carried out on an Agilent C18 column (Pursuit XRs C18, 150 × 2.0 mm, 5 µm) at room temperature with a flow rate of 0.4 ml min⁻¹. A gradient elution method employing two mobile phases was utilized. Mobile phase A consisted of 98% $H_2O$ and 2% acetonitrile with 0.1% formic acid, while mobile phase B comprised 98%

acetonitrile and 2% $H_2O$ with 0.1% formic acid. The gradient started with 2% B for the initial 0.5 min, then increased to 90% B over 8 min, held at 90% B for 4.5 min, and finally returned to the initial condition within 0.5 min. The column was re-equilibrated for 9 min with the initial condition before the next injection. The injection volume was 3 µl. Multiple Reaction Monitoring (MRM) in negative mode was employed for the assay of short-chain fatty acids. The MS/MS parameters were set as follows: curtain gas at 35 psi, source temperature at 600 °C, Gas 1 at 55 psi, Gas 2 at 55 psi, CAD at 10, Ion spray voltage at −4500 V, EP at −10 V, and CXP at -14. The MRM ion transitions for propionate, $[^{13}C_3]$propionate, D5 propionate, acetate, D5 acetate, butyrate, D7 butyrate, pentanoate, D9 pentanoate, hexanoate, and D11 hexanoate were 208/165, 211/167, 216/170, 194/151, 199/155, 222/179, 229/186, 236/193, 245/202, 250/207, and 261/218, respectively, with DP and CE set at −85 V and −18 V for propionate; −70 V and −18 V for acetate; −90 V and −19 V for butyrate; −94 V and −19 V for pentanoate; and −107 V and −22.7 V for hexanoate.

### Plasma ammonia assay

A modified LC-MS method was utilized to quantify ammonia in mouse plasma[63]. A 10 µl solution of $^{15}NH_4Cl$ (0.05 mM) served as the internal standard and was added to 20 µl of plasma samples. Plasma proteins were precipitated by the addition of 120 µl of methanol followed by vortexing. After centrifugation for 15 min at $12,000 \times g$, a 120 µl supernatant was mixed with 100 µl each of solution 1 (containing 100 mM phenol and 50 mg l$^{-1}$ sodium nitroprusside) and solution 2 (comprising 0.38 M dibasic sodium phosphate, 125 mM NaOH, and 1% sodium hypochlorite with available chlorine ranging from 10 to 15%). The mixture was then incubated at 37 °C for 40 min prior to LC-MS analysis.

For the ammonia phenol derivative assay, an LC-Q-Exactive$^+$-Orbitrap-MS instrument was employed. The Vanquish Binary Pump delivered a gradient elution consisting of two mobile phases at a flow rate of 0.4 ml min$^{-1}$. Mobile phase A comprised 98% $H_2O$ and 2% acetonitrile with 0.1% formic acid, while mobile phase B consisted of 98% acetonitrile and 2% $H_2O$ with 0.1% formic acid. The column used was an Agilent C18 column (Pursuit XRs C18, $100 \times 2.0$ mm, 5 µm) with a guard column, maintained at 45 °C in a column oven compartment. The autosampler temperature was set to 5 °C, with an injection volume of 5 µl. The entire analytical run lasted 18 min, with the gradient starting at 40% mobile phase B for 1.5 min, increasing to 90% mobile phase B within 4.5 min, maintaining 90% B for 4 min, then returning to 40% mobile phase B and equilibrating for 7.5 min before the next injection.

The Q-Exactive$^+$-Orbitrap-MS, equipped with a HESI probe, was configured with the following parameters: a heat temperature of 425 °C, sheath gas at 30, auxiliary gas at 13, sweep gas at 3, and a spray voltage of 3.5 kV in negative mode. Additionally, a capillary temperature of 320 °C and an S-lens setting of 45 were employed. A full m/z scan range was set from 60 to 900, with a resolution of 70,000 at m/z 200. The maximum injection time (max IT) was set at 200 ms, with the automated gain control (AGC) targeted at $3 \times 10^6$ ions. The unlabeled and $^{15}N$ labeled ammonia phenol derivatives were detected at m/z values of 198.0559 and 199.0530, respectively.

### Statistics and reproducibility

Statistical differences were determined by one-way analysis of variance (ANOVA) followed by Dunnett post-hoc test using Prism (GraphPad) software. The student's $t$ test was performed when two groups were compared and $P$ value was calculated in one-tailed. The number of replicates is defined in the figure legends.

Measured mass isotopologues distributions expressed as mol percent were corrected for natural enrichment[59,64]. M0, M1, M2, …, Mn, represents the n number of heavy atoms in a molecule.

The average carbon labeling of a metabolite is calculated with the following formula.

$$\text{Average carbon labeling of metabolite} = (m1 \times 1 + m2 \times 2 + m3 \times 3 + \dots + mn \times n)/n.$$ mn represents the enrichment of isotopomer with n number of heavy atoms in a molecule.

### Reporting summary

Further information on research design is available in the Nature Portfolio Reporting Summary linked to this article.

### Data availability

All data supporting the findings of this study are included in the article and its Supplementary information. Numerical source data for the graphs in the manuscript are available in Supplementary Data 1. Other data that support the findings of this study are available from the corresponding author upon request.

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

## Acknowledgements

This work is supported by the Propionic Acidemia Foundation award to G.F.Z., NIH R01 AA030026 to X.C. and G.F.Z. The authors express gratitude to Dr. Michael A. Barry (Mayo Clinic) for providing the *Pcca*$^{-/-}$(A138T) mice breeders for this study.

## Author contributions

W.H. and H.M. performed all experiments. W.H., H.M., D.K., T.K., X.C., and G.F.Z. participated data analysis, manuscript writing, and editing. All authors read and approved the manuscript in its final form.

## Competing interests

The authors declare no competing interests.
