## [Peer Review File · Communications Biology]

Reviewers' comments:

Reviewer #1 (Remarks to the Author):

"overnight fasting alleviates metabolic stress in mice with PCC mutation" is an interesting article which explores the impact of feeds in mice. It demonstrates that an overnight fast, in the Pcca (A138T) mice that there is enhanced acetyl-CoA production (in terms of measure by ratio) likely due to decreased gut production of propionyl-CoA.

In general, I do not have any overall significant concerns. However, I would like to caution the authors in terms of the clinical implications of this observation. This particular model system is based on less severe phenotype. It however is an intriguing observation. Seldom do we consider biochemical disorders with the observation of the whole organism. The muscle is being recognized in the propionate pathway disorder community as a significant biological contributor to stability.

Some of the tracer studies were done using muscle and other done using liver and since there is significant cross talk between these two important organs, I would have liked to see more discussion about how the observations in this paper illustrate some of that cross talk.

Minor question:

In the section about carbon flux, there is statement, "liver PCC activity remained unchanged...." can you finish it with "in the Pcca-/- (A138T) mice"? And is this true with the wild type controls or is inherent to having PCC deficiency?

In the discussion, second line, probably don't want antibodies, probably want it to say Antibiotics

Reviewer #2 (Remarks to the Author):

This is a very interesting study presented by He et al, that attempts to throw light on the pathophysiology of the propionic acidemia (PA), more specifically regarding a controversial issue of the metabolic response to fasting in mice with PA. Although we should be very careful interpreting the results and change the recommendations for the clinical practice, there are very interesting findings that surely deserve further investigations. The conclusions are original and the results might have an immediate influence on patients with PA.

To be sure, in the PA guidelines of 2021 (doi: 10.1002/jimd.12370), prolonged fasting periods are leading to catabolism and should be avoided to prevent metabolic instability with a moderate Quality of evidence and a strong strength of recommendation (supported by experience-based medical knowledge). Therefore, the patients or caregivers are specifically and strongly instructed to avoid fasting.

Major concerns:

1. The first is the enhanced fatty acids oxidation (FAO) during fasting. The elevation of BHB is obvious and only reflects the normal FAO pathway, which by the way is one of the few metabolic pathways that works normally in PA, as demonstrated by the authors results; the same guidelines mentioned above (doi: 10.1002/jimd.12370) recommend determining urine ketones each clinical visit as a marker for

catabolism and metabolic instability.

The surprise is when some of the PA metabolic parameters tend to improve, basically C2/C3 ratio and methylcitrate in plasma and organs.

2. The glucose production interpretation is more confusing and not totally convincing in my opinion. On the one hand, the authors state that glucose production during fasting is significantly lower in PA mice (lines 164-166, Figure 2G). Then they compare the glucose production of the PA mice in fast and in fed mice; they observe a higher glucose production in fasting mice, interpreting this finding as an increased carbon flux from propionyl CoA (lines 300-303), determined by the higher labeled glucose derived from ¹³C₃ propionate. In my opinion it is difficult to demonstrate that the glucose comes from propionyl CoA and TCA intermediates, considering that this is the main pathway affected in PA; to the note, I don't think that the propionyl CoA carboxylase activity is worth measuring in PA genetic mouse model, it will be the same, conditioned by the genetic mutation. There are other sources of glucose in the process of gluconeogenesis, such as glycerol or lactate. The source of glycerol is the lipolysis, needed to produce fatty acids for the enhanced FAO in fasting mice, as determined by the authors. The high lactate on the other side, very usual in PA metabolic crisis, might be responsible for the elevated pyruvate observed in fasting PA mice. Another marker for the gluconeogenesis, as well as pyruvate/lactate, is the alanine (Ala). Ala is usually low in plasma PA patients despite the high lactate, and I suggest these references: doi: 10.1007/s00726-022-03128-6, doi: 10.1016/j.ymgme.2005.11.016. This suggests deficient anaplerosis of the TCA cycle as explained in doi: 10.1007/s00726-022-03128-6.

This is one of my major concerns. Have the plasma alanine or lactate levels been determined, together with the plasma pyruvate? Consider presenting the results in Figure 7. I think it would be very useful for the discussion. The supplemental Figure 6 presents alanine and lactate in organs, I think it is more difficult to interpretate, considering the metabolic particularities of each organ.

I think in the Figure 7, PCC activity should be removed, as I would not expect any significant changes.

Also, the citrate levels might be removed, considering that there were no significant changes.

In the abstract, lines 38-39 should be reformulated to be more cautious about the gluconeogenesis from the propionyl CoA and the propionyl CoA activity that is not expected to change through any metabolic intervention. Also the lines 366-3367 should be reformulated in the same way, I don't think there is sufficient data to consider that glucose comes from propionate, or if the authors consider so, they should provide a more convincing explanation.

3. Microbiome propionate production is a very interesting finding that certainly deserves further investigation. Microbiome could be a major propionate source and, in my opinion, could be responsible for most of the most metabolic improvement.

4. Regarding the BCAA metabolism. An important observation of the authors is that the BCAT expression is prominent in the muscle (lines 224-225) and that the BCAT activity is increasing in fasting; still the C3 production does not increase significantly. Apparently during fasting there is an increase plasma levels of glutamate, isoleucine and leucine (Figure 4), with no significant increase in valine, KIV, 3HIB and C3 (Figure 5). Therefore, I don't see how the authors conclusions that the BCAA metabolism exhibited and increase in the initial step (lines 237-238) can be sustained. The isoleucine and leucine plasma levels can increase during important ketosis, through the generation of acetyl CoA (see reference doi: 10.1007/s00726-022-03128-6).

The plasma glutamate is an unstable metabolite, as it continuously converts to glutamine (Gln). The Gln is a major metabolic player in PA physiopathology (references: doi: 10.1007/s00726-022-03128-6, doi:

10.1016/j.ymgme.2005.11.016). As the authors depicted in the Figure 4A, the transamination of Val + 2-KG forms glutamate + KIV; then the glutamate forms glutamine, that is the most abundant plasma, with multiple physiologic roles (doi: 10.1007/s00726-022-03128-6). Have plasma Gln been measured? I think it would be useful to see if the N of the glutamine comes from the BCAA and which ones. Low glutamine levels are related to deficient anaplerosis of the TCA cycle (doi: 10.1007/s00726-022-03128-6). I think the plasma glutamine would be more useful than the plasma glutamate to be represented in the graphic.

5. Another major concern: has the plasma ammonia been measured together with the beta-OH butyrate? It would be useful to represent the plasma ammonia, as it is the major parameter of metabolic control in patients with PA, and we cannot conclude that the fasting improves the metabolic profile without taking into account the ammonia.

Minor concerns:

- Line 315. There is no treatment with "antibodies" in PA, I think the authors meant "antibiotics".
- The fasting group underwent a 23-24 hours fasting (line 408, Methods). To my understanding this is larger than the overnight fasting. Can the authors clarify? Another question is the age of the mice; I think it's important that authors clarify if there are newborns or adults mice, considering the peculiarities of the newborn metabolic pathways and the vulnerability for metabolic decompensation at this specific age group.
- What exactly is the meaning of "increased propionyl CoA" catabolism in line 265, Discussion? Please consider that the only metabolic pathway of the propionyl CoA is through the propionyl CoA carboxylase. The gluconeogenesis from propionyl CoA also implies the entry in the TCA cycle, that is impaired. As I mentioned above, the enhanced gluconeogenesis might come from the glycerol released in lipolysis or from lactate.

Reviewer #3 (Remarks to the Author):

This article describes the metabolic consequences of fasting in a hypomorphic transgenic mouse model of PCCA type propionic acidemia (PA) using a variety of methods including metabolites analyses, stable isotope studies and enzyme activity measurements. The authors found that fasting improved the metabolic abnormalities in blood and other tissues and suggest this as a potential therapeutic avenue to explore. The paper is well written, and authors are experts in this area with several other publications on propionic acidemia and this publication is likely to be of interest to the metabolic community. However, there are several comments to address before publication.

Major comments

1. One major caveat in this paper is the fact that fasting by definition includes a low protein intake and decreased propiogenic load, and so the decreased metabolites are not surprising. It is difficult therefore to separate fasting from low protein intake. This is not explicitly discussed in the paper. The authors

could consider comparison of mice on a low to no protein diet analogous to a sick day diet in a human which is routinely used in the treatment of patients with PA. This may not be in the scope of this project to at this point perform a separate arm of the experiments, but would be helpful to dissect out the consequences of fasting vs protein restriction only and strengthen the conclusions. At minimum please comment on this in detail in the manuscript and discuss in the discussion section as a limitation.

2. This mouse model has been used in several publications, including by the current authors and other groups but no context is provided in the manuscript and original paper was not referenced. In order to interpret the conclusions of this study, it is necessary to have background information on the mouse model used, specifically that this is a hypomorphic transgenic mouse overexpressing p.A138T that recapitulates the mild end of the spectrum of propionic acidemia, given its long term survival, fertility, etc PMID: 23648696 although does display the metabolic abnormalities in PA. Please add to the introduction and discussion (as a limitation) background about the mouse model used in this study and the associated references to highlight that it recapitulates the mild end of the PA spectrum with extremely mild biochemical as well as clinical phenotype, which is not representative of the majority of patients with PA. This is essential given some of the conclusions in this paper which state that fasting could be helpful.

3. The flow of the publication may be improved by moving figure 2 to the beginning of the paper as it compares the mutant mice to wild type and has some baseline values before studying the fasting state.

4. The authors made significant efforts to study the contribution of gut metabolites by the portal vein. It is unclear how they determined that these metabolites were exclusively derived from gut bacteria. Please explain this further how propionate from the portal vein circulation derives exclusively from the microbiome, instead of the diet from low protein load due to fasting. Were any additional studies performed for example genomic sequencing to confirm that the propiogenic bacteria were decreased?

5. Figure 5-7 isotope studies need more detailed figure legends to document all of the abbreviations used. There is too much information, too many graphs please consider moving some graphs to supplemental figures and keep figures that highlight the take home message. The figure schematics although helpful, are not very clear and don't include all of the detailed abbreviations and could be improved for readability, especially for readers that are not metabolic experts. This reviewer familiar with PA, mouse models and isotopes had a difficult time following these figures.

6. The conclusions of this study that fasting may be used a treatment for PA - "challenge traditional recommendations for PA patients and suggest the potential metabolic benefit of fasting" based on this study in a hypomorphic mouse model and caveat above #1 is somewhat concerning given current PA guidelines and decades of research in PA in humans (Fornly et al 2021 Recommendation #6). We would highly recommend reviewing the text of these conclusions with a metabolic physician familiar with propionic acidemia to make sure the limitations of this study are highlighted.

Minor comments

1. Please edit the title to specify the gene under study in this mouse model. There is no PCC gene. For example "overnight fasting alleviates metabolic stress in mice with propionyl CoA carboxylase deficiency due to PCCA". Also fasting is longer than overnight correct 23 hours ? Alleviates metabolic stress is very general consider more specific language to describe the findings in this manuscript.
2. The abstract, introduction and discussion and contain inaccuracies for a genetics audience "mutations in the gene", "PCC gene mutations". Please add both PCCA and PCCB genes to the introduction and accurately describe the genetics of propionic acidemia
3. Line 113 refers to a statement that overnight fasting alleviated the overall metabolic stresses in the mice. This is somewhat of an overstatement. Please edit.
4. It is surprising that the ACUC approved 23 hours of fasting in mice as it is a long period of time. Was there any additional monitoring of the mice? If so please describe.
5. Figure 1 please note that there is no figure legend for panel H please add
6. Figure 1 BCDEF appears that these figures contain the mutant mice only although it is not well documented in the figures nor legend. The data on the wild type controls is not presented please comment on the effects of fasting in the wild type controls.
7. Please comment on why the ratio of C2 to C3 Figure 1H was used. In PA/MMA/cbl newborn screening the opposite ratio C3 to C2 is used. This may be somewhat confusing to a metabolic audience as an improvement in the C2 to C3 ratio is shown as increased. We often think of metabolic improvement and a decrease in metabolites (and ratios in the case of C3/C2).
8. Figure 1J please describe how methylcitrate was measured, and the units presented. In other publications often nmol/L, or umol/L are used so it is difficult to compare these values to other publications even though it appears significantly decreased and fasting mice. It's interesting to document that methylcitrate wasn't decreased in all tissues, particularly the heart and brain which are organs which are affected by PA, please comment on this in the discussion.
9. In figure 2E and F please explain the Y axis what is the fold change comparator ??
10. Supplemental 2b it doesn't appear that C17 heptadecanoic acid was measured in this study, a figure may not be required.
11. Propionic acid is reportedly notoriously difficult to measure due to instability, so is your measurement Propionic acid or a derivative 3 hydroxypropionate etc please specify
12. Figure 3G is described in the text line 203 stating that propionate emerges the most prominent metabolite exhibiting a substantial decrease this statement should be revised to state that is the most statistically significant based on the P value

13. Figure 5 is difficult to follow as it doesn't describe M0, the Y axes are similar, figure legend is not very detailed. In the text please comment specifically on which figure that shows 3HIB (line 232). It could be helpful to show a figure with the larger BCAA pathway to orient the reader to the larger picture.

14. Line 241 metabolic substrate of propionyl CoA or propionyl CoA carboxylase?

15. Similar comment to Figure 5, difficult to follow. There are several things that are not well described M0, M5. Perhaps some graphs could be moved to supplementals and focus on showing panels that are important/essential to the results and described in the text.

16. Same comment for Figure 7.

17. Figure 7H what was the PCC activity in the mutant mice compared to controls??

18. Line 306 I think you are referring to Suppl Fig 7 not 6 here

19. Line 328 "correction of metabolic rewiring" is an overstatement

Re: COMMSBIO-23-4733-T

Point-to-point response

Reviewers' comments:

Reviewer #1 (Remarks to the Author):

1. "overnight fasting alleviates metabolic stress in mice with PCC mutation" is an interesting article which explores the impact of feeds in mice. It demonstrates that an overnight fast, in the *Pcca* (A138T) mice that there is enhanced acetyl-CoA production (in terms of measure by ratio) likely due to decreased gut production of propionyl-CoA.

Response: Thank you for the generally positive feedback on this work.

2. In general, I do not have any overall significant concerns. However, I would like to caution the authors in terms of the clinical implications of this observation. This particular model system is based on less severe phenotype. It however is an intriguing observation. Seldom do we consider biochemical disorders with the observation of the whole organism. The muscle is being recognized in the propionate pathway disorder community as a significant biological contributor to stability.

Response: Thank you for the suggestive comments on this work. We acknowledge the need for extreme caution when comes to clinical implications. Hence, we emphasize the limitations of this study, particularly regarding its translation into clinical applications. We intend to delve into these limitations in greater detail in future research endeavors. It's acknowledged that the *Pcca*^{-/-}(A138T) mouse model represents a less severe phenotype (this limitation has been added into the discussion in the revised manuscript), yet it exhibits a robust biochemical phenotype, as evidenced by plasma propionyl-carnitine/propionate levels exceeding 30-fold compared to wild type (see below data).

This model has proven valuable in studies related to gene therapy, dual mRNA therapy, and various drug treatments, as referenced (PMID: 25046265, PMID: 23648696, PMID: 33087718, PMID: 34524863, and PMID: 36251252).

We concur with the reviewer's observation regarding the significant role of muscle in propionyl-CoA metabolism, particularly given its high branched-chain amino acid transaminase (BCAT) activity, where

valine/isoleucine serve as substrates for propionyl-CoA. Liver also plays pivotal roles in branched-chain amino acid catabolism due to the transfer of branched-chain keto acids from muscle to liver and the liver's high branched-chain keto acid dehydrogenase (BCKDH) activity, as cited (PMID: 29779826 and PMID: 27408778). Moreover, the liver exhibits a higher capacity for propionate and odd-chain fatty acid metabolism. It serves as the primary site where propionate derived from the microbiome is efficiently taken up from portal vein and metabolized (~100% uptake by the liver, PMID: 10600790). Lastly, the liver's predominant role in propionyl-CoA metabolism is strongly evidenced by the effective treatment of propionic acidemia (PA) through liver transplantation in patients, as supported by references (PMID: 35068050 and PMID: 31715057).

3. Some of the tracer studies were done using muscle and other done using liver and since there is significant cross talk between these two important organs, I would have liked to see more discussion about how the observations in this paper illustrate some of that cross talk.

Response: We agree with the reviewer's comments regarding the roles of tissues in propionyl-CoA metabolism. All tracer studies were conducted in whole animals, and the tracing data were presented for various organs including muscle and liver in both main and supplemental figures.

The cross-talk between important metabolic organs such as the liver and muscle was clearly demonstrated in this study as well. For example, in the case of stable isotope-labeled valine, it is known that BCAT activity is low in the liver. This is evidenced by the lowest levels of M5 KIV in the liver (Supplemental Figure 5E). However, the levels of further downstream metabolites (M4 3HIB and M3 C3 AC) are much higher in the liver than in muscle (Supplemental Figures 5H and 5K). This confirms that valine metabolism could originate in muscle, with metabolic intermediates (3HIB, C3 AC, and others) being transferred to the liver for further catabolism. This observation aligns with findings reported in the literatures, as mentioned in our response to the reviewer.

Another example is the activated threonine metabolism in the liver. The activated SDS activity in the liver during fasting is >10 times higher than in muscle (Figure 6G and Supplemental Figure 6E). However, the labeling and levels of 2HB in the liver and muscle are not significantly different (Supplemental Figures 6I and 6K). This further demonstrates that metabolic intermediates could undergo cross-talk between organs.

We have also included additional discussions on cross-talk in the manuscript.

Minor question:

1. In the section about carbon flux, there is statement, "liver PCC activity remained unchanged...." can you finish it with "in the *Pcca*^{-/-} (A138T) mice"? And is this true with the wild type controls or is inherent to having PCC deficiency?

Response: We added "in the *Pcca*^{-/-} (A138T) mice" in the end of sentence. We conducted PCC activity change in wild type mice after 23-hour fasting. The results are shown in the below and added into Supplemental Figure 8I. Enzyme activity assay in wild type mice also confirmed the results in *Pcca*^{-/-} (A138T) mice that fasting does not change PCC acidity in both wild type and mutated mice.

2. In the discussion, second line, probably don't want antibodies, probably want it to say Antibiotics

Response: Thank you for catching our typo. We corrected it to “antibiotics”.

Reviewer #2 (Remarks to the Author):

1. This is a very interesting study presented by He et al, that attempts to throw light on the pathophysiology of the propionic acidemia (PA), more specifically regarding a controversial issue of the metabolic response to fasting in mice with PA. Although we should be very careful interpreting the results and change the recommendations for the clinical practice, there are very interesting findings that surely deserve further investigations. The conclusions are original and the results might have an immediate influence on patients with PA.

Response: Authors greatly appreciate the reviewer's recognition of the originality and significance of the findings presented in this work.

2. To be sure, in the PA guidelines of 2021 (doi: 10.1002/jimd.12370), prolonged fasting periods are leading to catabolism and should be avoided to prevent metabolic instability with a moderate Quality of evidence and a strong strength of recommendation (supported by experience-based medical knowledge). Therefore, the patients or caregivers are specifically and strongly instructed to avoid fasting.

Response: Thank you for bringing attention to the reference regarding PA guidelines. We have duly acknowledged and integrated this citation into the manuscript. The contradictory findings presented in this work contribute to the novelty of the manuscript. However, it is important to highlight that this study marks the initial investigation into the effects of fasting on metabolic changes in a mouse model of PA. We exercise caution regarding its direct clinical implications, as outlined in the limitations section in the revised manuscript. As noted by the reviewer, this intriguing discovery calls for further exploration, particularly in human subjects. Additional limitations and caution have been incorporated into the discussion section of the revised manuscript.

Major concerns:

1. The first is the enhanced fatty acids oxidation (FAO) during fasting. The elevation of BHB is obvious and only reflects the normal FAO pathway, which by the way is one of the few metabolic pathways that works normally in PA, as demonstrated by the authors results; the same guidelines mentioned above (doi: 10.1002/jimd.12370) recommend determining urine ketones each clinical visit as a marker for

catabolism and metabolic instability.

The surprise is when some of the PA metabolic parameters tend to improve, basically C2/C3 ratio and methylcitrate in plasma and organs.

Response: As highlighted by this reviewer, fasting is anticipated to elevate fatty acid oxidation (FAO). In this study, BHB, a ketone body, specifically denotes 3-hydroxybutyrate derived from fatty acid oxidation. Nevertheless, the terminology surrounding ketones or ketone bodies in PA field can be perplexing. For instance, numerous studies have documented urinary ketone bodies such as 3-OH isovaleric acid, 3-OH propionate, and 3-pentanone (examples include PMID: 15164333, PMID: 486715). Notably, these ketones, associated with propionyl-CoA metabolism not fatty acid oxidation, undergo substantial elevation during metabolic decompensation in PA. However, BHB levels remain within normal ranges in PA cases without metabolic decompensation (PMID: 32143654). In the presented mouse model, BHB levels in *Pcca*^{-/-} (A138T) mice are even lower than those in WT mice (Figure 2E). It is imperative to note that our experiments were conducted using *Pcca*^{-/-} (A138T) mice without metabolic decompensation. Furthermore, a reported study (DOI: 10.1016/j.ymgme.2005.11.016) indicated no correlation between urinary BHB and plasma ammonia levels in PA patients, as underscored by this reviewer in the subsequent comment (comment 2). Conversely, 3-OH propionate, propionylglycine, and methylcitrate in urine exhibited a positive correlation with plasma ammonia. Hence, there is a clear necessity to elucidate the identity of ketones documented in previous reports on PA.

The enhanced fatty acid oxidation effectively augments acetyl-CoA production, thereby ameliorating the C2/C3 ratio imbalance. The corrected C2/C3 ratio restores the metabolic alterations induced by the aberrant C2/C3 ratio. This is evidenced by the elevated levels of BHB (Figure 1F and Supplemental Figure 1A) and acylcarnitines (and Figure 2H), particularly in the liver. The substantial increase in C2 levels derived from fatty acid oxidation in the liver is supported by the data showing in Supplemental Figure 1E.

2. The glucose production interpretation is more confusing and not totally convincing in my opinion. On the one hand, the authors state that glucose production during fasting is significantly lower in PA mice (lines 164-166, Figure 2G). Then they compare the glucose production of the PA mice in fast and in fed mice; they observe a higher glucose production in fasting mice, interpreting this finding as an increased carbon flux from propionyl CoA (lines 300-303), determined by the higher labeled glucose derived from ¹³C₃ propionate.

Response: We apologize for any confusion that may have arisen from the manuscript. As pointed out by this reviewer, two distinct comparisons are being made. Firstly, the comparison involves the gluconeogenesis capacity between wild-type and PA mice following a 5-hour fasting period, assessed using the D2 glucose approach (PMID: 27651111). Reduced propionyl-CoA carboxylase (PCC) activity leads to decreased gluconeogenesis in PA mice compared to wild-type mice (Figure 2G). Secondly, the comparison relates to gluconeogenesis in PA mice under both fed and fasting conditions. Despite gluconeogenesis being attenuated in PA mice, it is not entirely abolished. Consequently, gluconeogenesis remains significantly increased after a 23-hour fasting period compared to the fed condition (Figure 7D). Fasting, in contrast to the fed condition, augments carbon flux from [¹³C₃]propionate to glucose synthesis.

3. In my opinion it is difficult to demonstrate that the glucose comes from propionyl CoA and TCA intermediates, considering that this is the main pathways affected in PA; to the note, I don't think that the propionyl CoA carboxylase activity is worth measuring in PA genetic mouse model, it will be the

same, conditioned by the genetic mutation. There are other sources of glucose in the process of gluconeogenesis, such as glycerol or lactate. The source of glycerol is the lipolysis, needed to produce fatty acids for the enhanced FAO in fasting mice, as determined by the authors. The high lactate on the other side, very usual in PA metabolic crisis, might be responsible for the elevated pyruvate observed in fasting PA mice. Another marker for the gluconeogenesis, as well as pyruvate/lactate, is the alanine (Ala). Ala is usually low in plasma PA patients despite the high lactate, and I suggest these references: doi: 10.1007/s00726-022-03128-6, doi: 10.1016/j.ymgme.2005.11.016. This suggests deficient anaplerosis of the TCA cycle as explained in doi: 10.1007/s00726-022-03128-6. This is one of my major concerns. Have the plasma alanine or lactate levels been determined, together with the plasma pyruvate? Consider presenting the results in Figure 7. I think it would be very useful for the discussion. The supplemental Figure 6 presents alanine and lactate in organs, I think it is more difficult to interpretate, considering the metabolic particularities of each organ. I think in the Figure 7, PCC activity should be removed, as I would not expect any significant changes. Also, the citrate levels might be removed, considering that there were no significant changes. In the abstract, lines 38-39 should be reformulated to be more cautious about the gluconeogenesis from the propionyl CoA and the propionyl CoA activity that is not expected to change through any metabolic intervention. Also the lines 366-3367 should be reformulated in the same way, I don't think there is sufficient data to consider that glucose comes from propionate, or if the authors consider so, they should provide a more convincing explanation.

Response: Thank you for the insightful feedback on our discussion and data interpretation regarding gluconeogenesis. We value the opportunity to provide a more compelling explanation as an alternative perspective. Our study encompasses a comprehensive array of stable isotope tracing experiments, incorporating numerous technical terms associated with stable isotopes. The interpretation of stable isotope labeling data (kinetic change of metabolism) could be more complicated than the conventional metabolic data (snapshot of static concentration data).

In our investigation, the incorporation of ^{13}C into metabolites elucidates the metabolic pathways through which ^{13}C from labeled substrate travels. Specifically, in the [$^{13}\text{C}_3$]propionate tracing experiment, the presence of ^{13}C in glucose indicates that the newly synthesized glucose contains ^{13}C labeled carbon derived from ^{13}C -labeled propionate. This experimental approach directly substantiates the occurrence of gluconeogenesis from [$^{13}\text{C}_3$]propionate, even in the context of reduced PCC activity observed in $Pcca^{-/-}$ (A138T) mice.

It is important to note that while PCC activity is reduced in $Pcca^{-/-}$ (A138T) mice, it is not entirely abolished, as evidenced by our published work (PMID: 34635437) and the data on PCC activity in various tissues from both wild-type and $Pcca^{-/-}$ (A138T) mice, provided below for reference.

Although PCC was mutated in our mouse model, its expression and activity could still be regulated. This was demonstrated in papers from us and others, where we found that PCC expression and activity vary depending on the tissues and genders despite sharing the same mutated gene (PMID: 34635437 and PMID: 34524863). One of the goals of our PA project is to identify the regulatory mechanism of PCC. We found that the incorporation of ^{13}C into newly synthesized glucose is increased in *Pcca*^{-/-}(A138T) mice during fasting (see Figure 7D). Naturally, this finding prompts the question of whether PCC activity is influenced by fasting. To address this inquiry, we measured and presented PCC activity. However, to simplify the Figure, we removed the PCC activity data to supplemental Figure 8H.

As suggested by the reviewer, we quantified alanine, glutamine, glutamate, glycerol, pyruvate, and lactate in plasma from fed and fasted mice (see figures below, they were added into Supplemental Figures 8A-8E). Most of these metabolites tended to decrease during fasting, except for glycerol, which showed a trending increase (although not significant) possibly due to lipolysis, as noted by the reviewer. The decrease in other metabolites could be attributed to anaplerosis for gluconeogenesis, as suggested by the reviewer. If the increase in gluconeogenesis was solely due to the increased flux of alanine/glycerol, we would not have observed the increase in glucose labeling during fasting (Figure 7D). Furthermore, if anaplerosis was solely from glutamine or other unlabeled substrates but not from ^{13}C propionate, then the ^{13}C citrate labeling would have decreased during fasting. However, citrate average C labeling (not citrate levels) was maintained (Figure 7C), despite anaplerosis from other unlabeled substrates including glutamine increasing during fasting. This demonstrates that the metabolic flux of propionyl-CoA to the TCA cycle increased, even though PCC activity remained. Furthermore, this provides direct and clear evidence demonstrating that the carbon flux from [$^{13}\text{C}_3$]propionate to glucose synthesis was increased during fasting.

The references cited by this reviewer and the discussion of anaplerosis from other substrates have been incorporated into the revised manuscript.

4. Microbiome propionate production is a very interesting finding that certainly deserves further investigation. Microbiome could be a major propionate source and, in my opinion, could be responsible for most of the most metabolic improvement.

Response. We concur with the reviewer regarding the intriguing data concerning propionate derived from the microbiome. We will pursue this finding further to elucidate the precise mechanism. Additionally, we agree with the reviewer regarding the potential metabolic improvements resulting from the reduction of propionate. Furthermore, we have data demonstrating that the microbiome serves as a significant source of propionate (refer to our response to major comment 4 from reviewer 3).

5. Regarding the BCAA metabolism. An important observation of the authors is that the BCAT expression is prominent in the muscle (lines 224-225) and that the BCAT activity is increasing in fasting; still the C3 production does not increase significantly. Apparently during fasting there is an increase plasma levels

of glutamate, isoleucine and leucine (Figure 4), with no significant increase in valine, KIV, 3HIB and C3 (Figure 5).

Therefore, I don't see how the authors conclusions that the BCAA metabolism exhibited and increase in the initial step (lines 237-238) can be sustained.

Response. Apologies for any confusion regarding our stable isotope labeling data. In Figure 4, we presented stable isotope labeling data of glutamate (^{15}N), isoleucine (^{15}N), and leucine (^{15}N), which serve to demonstrate the BCAT activity in the stable isotope-based metabolic flux study. It's important to note that these data represent labeling, not concentrations. The labeling data (Figures 4B-4D) aligns well with the results of direct BCAT enzyme activity assay, as depicted in Figure 4E.

6. The isoleucine and leucine plasma levels can increase during important ketosis, through the generation of acetyl CoA (see reference doi: 10.1007/s00726-022-03128-6).

Response: We appreciate the reviewer's insightful remark and the provided reference. We regret any confusion caused by the stable isotope labeling analysis data. In Figures 4C and 4D, we presented the ^{15}N labeling data obtained from ^{15}N valine through BCAT-mediated transamination. The heightened ^{15}N labeling observed in isoleucine and leucine from ^{15}N valine indicates an upregulation of BCAT activity during fasting, a finding corroborated by direct enzyme activity assay (Figure 4E). Notably, our stable isotope-based tracing study from ^{13}C labeled valine did not indicate an increase in the catabolism of branched-chain amino acids (Figure 5).

7. The plasma glutamate is an unstable metabolite, as it continuously converts to glutamine (Gln). The Gln is a major metabolic player in PA physiopathology (references: doi: 10.1007/s00726-022-03128-6, doi: 10.1016/j.ymgme.2005.11.016). As the authors depicted in the Figure 4A, the transamination of Val + 2-KG forms glutamate + KIV; then the glutamate forms glutamine, that is the most abundant plasma, with multiple physiologic roles (doi: 10.1007/s00726-022-03128-6). Have plasma Gln been measured? I think it would be useful to see if the N of the glutamine comes from the BCAA and which ones. Low glutamine levels are related to deficient anaplerosis of the TCA cycle (doi: 10.1007/s00726-022-03128-6). I think the plasma glutamine would be more useful than the plasma glutamate to be represented in the graphic.

Response. We concur with the reviewer's statement. We indeed measured the levels of glutamine/glutamate in plasma, as demonstrated in our response to comment No. 2 from this reviewer. Consistent with the reviewer's suggestion, we observed a decrease in glutamine levels in plasma from fasted *Pcca*^{-/-}(A138T) mice, likely attributable to deficient anaplerosis of the TCA cycle.

It's important to note that the data presented here are stable isotope labeling data, aimed at illustrating the increased BCAT-mediated transamination step. Glutamate serves as the direct product of BCAT (Figure 4A). The ^{15}N labeling of glutamate (M1), isoleucine (M1), and leucine (M1) reflects BCAT-mediated metabolic flux, a finding supported by the enzyme activity assay (Figure 4E).

Additionally, we measured M1 labeled glutamine (as shown in the figure below) as requested by this reviewer. Glutamine could be synthesized from glutamate. Interestingly, there was no significant difference in M1 glutamine labeling between fed and fasted plasma, despite an increase in M1 glutamate during fasting (Figure 4B). This discrepancy might be explained by energy demanding step of glutamine synthesis.

8. Another major concern: has the plasma ammonia been measured together with the beta-OH butyrate? It would be useful to represent the plasma ammonia, as it is the major parameter of metabolic control in patients with PA, and we cannot conclude that the fasting improves the metabolic profile without taking into account the ammonia.

Response: Thank you to this reviewer for bringing up this question. We conducted the assay of ammonia levels in plasma (see below data), and this new data has been included in the revised manuscript (Supplemental Figure 1I). We observed a reduction in ammonia levels in the fasting plasma, which supports the improved metabolic changes observed in *Pcca*^{-/-}(A138T) mice.

Minor concerns:

1. - Line 315. There is no treatment with "antibodies" in PA, I think the authors meant "antibiotics".

Response: Thank you for bringing the typo to our attention. It has been corrected in the revised manuscript.

2. - The fasting group underwent a 23-24 hours fasting (line 408, Methods). To my understanding this is larger than the overnight fasting. Can the authors clarify? Another question is the age of the mice; I think it's important that authors clarify if there are newborns or adults mice, considering the peculiarities of the newborn metabolic pathways and the vulnerability for metabolic decompensation at this specific age group.

Response: The authors acknowledge the oversight. The fasting regimen employed in our study spanned one day, which was corrected in the revised manuscript. Additionally, it should be noted that the mice utilized in this research were adult mice, and their ages (16-20 weeks old) have been included in the revised manuscript (Methods section: Mice tracing experiment with intraperitoneal injections).

3. - What exactly is the meaning of "increased propionyl CoA" catabolism in line 265, Discussion? Please consider that the only metabolic pathway of the propionyl CoA is through the propionyl CoA carboxylase. The gluconeogenesis from propionyl CoA also implies the entry in the TCA cycle, that is impaired. As I mentioned above, the enhanced gluconeogenesis might come from the glycerol released in lipolysis or from lactate.

Response: The enhanced propionyl-CoA catabolism indicates an elevated carbon flux from propionyl-CoA to glucose synthesis during fasting, as evidenced by Figures 7D and supplemental Figures 8J and 8K. In the revised manuscript, we have rephrased this to "increased propionyl-CoA flux to glucose synthesis" for clarity. For a comprehensive explanation, please see our response to Major Comment 2 from this reviewer.

Reviewer #3 (Remarks to the Author):

This article describes the metabolic consequences of fasting in a hypomorphic transgenic mouse model of PCCA type propionic acidemia (PA) using a variety of methods including metabolites analyses, stable isotope studies and enzyme activity measurements. The authors found that fasting improved the metabolic abnormalities in blood and other tissues and suggest this as a potential therapeutic avenue to explore. The paper is well written, and authors are experts in this area with several other publications on propionic acidemia and this publication is likely to be of interest to the metabolic community. However, there are several comments to address before publication.

Response: The authors are grateful for the overall positive feedback provided by this reviewer.

Major comments

1. One major caveat in this paper is the fact that fasting by definition includes a low protein intake and decreased propiogenic load, and so the decreased metabolites are not surprising. It is difficult therefore to separate fasting from low protein intake. This is not explicitly discussed in the paper. The authors could consider comparison of mice on a low to no protein diet analogous to a sick day diet in a human which is routinely used in the treatment of patients with PA. This may not be in the scope of this project to at this point perform a separate arm of the experiments, but would be helpful to dissect out the consequences of fasting vs protein restriction only and strengthen the conclusions. At minimum please comment on this in detail in the manuscript and discuss in the discussion section as a limitation.

Response. The reviewer has made a valid point and proposed an intriguing idea. We acknowledge that fasting and protein restriction diets share some similarities in terms of protein and amino acid intake. However, it's important to note that the purpose of a low protein diet is primarily to reduce protein and amino acid intake, thereby limiting their catabolism to propionyl-CoA. Fasting involves more than just protein restriction. The low calorie during fasting may promote protein breakdown and subsequent catabolism to propionyl-CoA, which is why fasting is generally not recommended in clinical practice.

Patients with PA often exhibit low levels of propiogenic amino acids due to protein restriction diets (PMID: 35098378 and PMID: 16406646). However, the profile of propiogenic amino acids in fasted Pcca-/- (A138T) mice differs. During fasting, isoleucine and threonine levels increase (see below data). While the protein restriction diet has been extensively studied and implemented clinically, comparing it to

fasting presents an interesting avenue for investigating the differences between low protein diets and fasting. However, we agree with the reviewer that this is beyond the scope of our current work. Nonetheless, we have included additional discussion on this topic in the revised manuscript.

Most importantly, we observed a significant reduction in propionate, a product of the gut microbiome, during fasting. In addition to the decrease in propionyl-CoA synthesis, there is also an increase in the carbon flux from propionyl-CoA to glucose. Therefore, fasting encompasses more than just dietary restriction.

2. This mouse model has been used in several publications, including by the current authors and other groups but no context is provided in the manuscript and original paper was not referenced. In order to interpret the conclusions of this study, it is necessary to have background information on the mouse model used, specifically that this is a hypomorphic transgenic mouse overexpressing p.A138T that recapitulates the mild end of the spectrum of propionic acidemia, given its long term survival, fertility, etc PMID: 23648696 although does display the metabolic abnormalities in PA. Please add to the introduction and discussion (as a limitation) background about the mouse model used in this study and the associated references to highlight that it recapitulates the mild end of the PA spectrum with extremely mild biochemical as well as clinical phenotype, which is not representative of the majority of patients with PA. This is essential given some of the conclusions in this paper which state that fasting could be helpful.

Response. Thank you for the helpful suggestions. We concur that incorporating background information on the mouse model will enhance the contextual understanding of this work. As advised by this reviewer, we have included the background of the mouse model in both the introduction and discussion sections.

3. The flow of the publication may be improved by moving figure 2 to the beginning of the paper as it compares the mutant mice to wild type and has some baseline values before studying the fasting state.

Response. Thank you for the suggestive comments. The wild-type (WT) data included in this study only serves to demonstrate that *Pcca*^{-/-}(A138T) mice rely more on fatty acid oxidation during fasting, despite having lower fat mass compared to WT mice (Figure 2B and Figure 2E). During fasting, the liver plays a crucial role in providing glucose through gluconeogenesis. The attenuated gluconeogenesis observed in *Pcca*^{-/-}(A138T) mice (Figure 2G) may lead the liver to prioritize ketone production through increased fatty acid oxidation (Figure 2F). Therefore, the primary focus of this work is not to compare genetic backgrounds but rather to investigate the effects of fasting.

The authors prefer to retain these data in Figure 2. However, if the reviewer insists, we can consider revising the figure order accordingly.

4. The authors made significant efforts to study the contribution of gut metabolites by the portal vein. It is unclear how they determined that these metabolites were exclusively derived from gut bacteria. Please explain this further how propionate from the portal vein circulation derives exclusively from the microbiome, instead of the diet from low protein load due to fasting. Were any additional studies performed for example genomic sequencing to confirm that the propiogenic bacteria were decreased?

Response. Thank you for this reviewer's critical comments. It has been established that short-chain fatty acids are derived from the microbiome (PMID: 10600790), and this reference has been cited in the revised manuscript.

In our other work, we measured the levels of short-chain fatty acids in the portal veins of control and germ-free mice (see figure below). Propionate levels were assessed in plasma withdrawn from various locations, including the hepatic vein (HV), renal vein (RV), and portal vein (PV), in both control and germ-free (GF) mice. In control mice, propionate levels were found to be highest in the portal vein. This pattern was consistent for other short-chain fatty acids ranging from C2 (acetate) to C5 (pentanoate). These findings align with existing literature and unequivocally demonstrate that short-chain fatty acids, including propionate, originate from the microbiome.

5. Figure 5-7 isotope studies need more detailed figure legends to document all of the abbreviations used. There is too much information, too many graphs please consider moving some graphs to supplemental figures and keep figures that highlight the take home message. The figure schematics although helpful, are not very clear and don't include all of the detailed abbreviations and could be improved for readability, especially for readers that are not metabolic experts. This reviewer familiar with PA, mouse models and isotopes had a difficult time following these figures.

Response: Thank you for your constructive comments. We removed some graphs in Figures 5-7 to supplemental figures. The revised figure schematics included all of the detailed abbreviation.

6. The conclusions of this study that fasting may be used a treatment for PA - "challenge traditional recommendations for PA patients and suggest the potential metabolic benefit of fasting" based on this study in a hypomorphic mouse model and caveat above #1 is somewhat concerning given current PA guidelines and decades of research in PA in humans (Fornly et al 2021 Recommendation #6). We would highly recommend reviewing the text of these conclusions with a metabolic physician familiar with propionic acidemia to make sure the limitations of this study are highlighted.

Response: We acknowledge the concerns raised by the reviewer and wish to clarify the primary goal of our study, which is not to advocate specific fasting regimens as a treatment for PA. Rather, the novelty lies in our findings, which contradict prevailing clinical recommendations, including those referenced by the reviewer. Our observations are substantiated by our tracing study, short-chain fatty acid production assay, and enzyme activity measurements.

The intriguing results underscore the need for researchers and clinicians to reevaluate the impact of fasting on PA, an area that has not been extensively studied despite its clinical endorsement.

Dr. Dwight Koeberl, a co-author of this study, is a physician-scientist specializing in clinical and biochemical genetics, particularly in the treatment of patients with inherited metabolic disorders such as PA. His expertise in metabolic medicine, specifically in PA, has been instrumental in our collaborative efforts. By interacting with Dr. Koeberl, we recognize the limitations of our research. These limitations were added into the revised manuscript (last two paragraphs).

- (1) This study employs mouse models, and while commonly utilized in disease research, discrepancies between mouse and human physiology may exist.
- (2) The *Pcca*^{-/-} (A138T) mouse model represents only one mutation associated with PA. Given the diverse spectrum of mutations in PA patients, the generalizability of our findings to all PA mutations remains uncertain.
- (3) The metabolic status of *Pcca*^{-/-} (A138T) mice in our research are not in metabolic decompensation state.
- (4) Further investigations are warranted to elucidate any potential clinical implications before definitive conclusions can be drawn.

Minor comments

1. Please edit the title to specify the gene under study in this mouse model. There is no PCC gene. For example "overnight fasting alleviates metabolic stress in mice with propionyl CoA carboxylase deficiency due to PCCA". Also fasting is longer than overnight correct 23 hours ? Alleviates metabolic stress is very general consider more specific language to describe the findings in this manuscript.

Response: We incorporated the suggestions from this reviewer and adjusted the title to " Fasting alleviates metabolic alterations in mice with propionyl-CoA carboxylase deficiency due to *Pcca* mutation." The metabolic alterations included C2 and C3 levels, C2 AC/C3 AC ratio, methylcitrate, methylcitrate/citrate ratio. Additionally, we included the decrease of ammonia levels in the fasting plasma. The improved C2/C3 ratio effectively reverses all these metabolic alterations triggered by the buildup of propionyl-CoA in *Pcca*^{-/-} (A138T) mice. To simplify the title within the 15-word limitation, we therefore change the title to "Fasting alleviates metabolic alterations in mice with propionyl-CoA carboxylase deficiency due to *Pcca* mutation".

2. The abstract, introduction and discussion and contain inaccuracies for a genetics audience "mutations in the gene", "PCC gene mutations". Please add both PCCA and PCCB genes to the introduction and accurately describe the genetics of propionic acidemia

Response: We made the corrections suggested by this reviewer.

3. Line 113 refers to a statement that overnight fasting alleviated the overall metabolic stresses in the mice. This is somewhat of an overstatement. Please edit.

Response: We revised the statement to “Surprisingly, a 23-hour fasting period was observed to alleviate certain metabolic alterations in $Pcca^{-/-}$ (A138T) mice”.

4. It is surprising that the ACUC approved 23 hours of fasting in mice as it is a long period of time. Was there any additional monitoring of the mice? If so please describe.

Response: Thanks for critical comments. Rodents deprived of food for more than 24 hours will express certain physiological and behavioral adaptations and needs IACUC additional approval. Our experiment was within 24 hours.

5. Figure 1 please note that there is no figure legend for panel H please add

Response: We have reordered the legends to maintain consistency with the arrangement of the figure panels.

6. Figure 1 BCDEF appears that these figures contain the mutant mice only although it is not well documented in the figures nor legend. The data on the wild type controls is not presented please comment on the effects of fasting in the wild type controls.

Response: We included the mouse information in the legend of Figure 1. Wild type control mice were not included in this study as it specifically examined the fasting effects on PA mice. The effects of fasting are well-documented in wild type animals and humans. Nonetheless, we conducted experiments with wild type mice, and the changes in their body composition are presented below. As expected, there was a significant decrease in fat mass observed in fasted mice, suggesting increased fatty acid oxidation.

7. Please comment on why the ratio of C2 to C3 Figure 1H was used. In PA/MMA/cbl newborn screening the opposite ratio C3 to C2 is used. This may be somewhat confusing to a metabolic audience as an improvement in the C2 to C3 ratio is shown as increased. We often think of metabolic improvement and a decrease in metabolites (and ratios in the case of C3/C2).

Response: Based on the reviewer's suggestion, we modified it to the C3/C2 ratio in Figure 1H.

8. Figure 1J please describe how methylcitrate was measured, and the units presented. In other publications often nmol/L, or umol/L are used so it is difficult to compare these values to other publications even though it appears significantly decreased and fasting mice. It's interesting to document that methylcitrate wasn't decreased in all tissues, particularly the heart and brain which are organs which are affected by PA, please comment on this in the discussion.

Response: The metabolites of organic acids and amino acids were measured by GC-MS, which was described in method section. We presented methylcitrate to μM units in Figure 1J. The alteration in methylcitrate levels is contingent upon the relative levels of propionyl-CoA and acetyl-CoA, as they compete with citrate synthetase to form either methylcitrate or citrate. Changes in propionate production from the microbiome (Figure 3B) and increased fatty acid oxidation to acetyl-CoA alter the ratio of acetyl-CoA and propionyl-CoA, particularly in the liver (Supplemental Figure 1F), as the liver is the primary organ exposed to propionate derived from the microbiome.

Regarding the heart, our previous research indicated that significant metabolic flux of propionyl-CoA to the TCA cycle in $Pcca^{-/}(A1398T)$ mice may be attributed to uneven expressions of mutated PCCA among tissues (PMID: 34635437). The unchanged and low levels of methylcitrate (Supplemental Figure 1G) could be attributed to relatively high PCC activity in the $Pcca^{-/}(A1398T)$ heart.

In the brain, methylcitrate levels are typically low (Supplemental Figure 1G). The slight, yet significant, increase in methylcitrate might suggest low acetyl-CoA production in the brain during fasting conditions. This is further supported by the increased citrate labeling from $^{13}\text{C}_3$ propionate (Supplemental Figure 7C). The increased citrate labeling during fasting indicated the potential the decreased acetyl-CoA flux to TCA cycle, if PCC activity was not changed (Supplemental Figure 7H). These have been incorporated into the discussion and warrant further investigation.

9. In figure 2E and F please explain the Y axis what is the fold change comparator ??

Response: The fold change was compared to the wild type, and this information has been included in the legend.

10. Supplemental 2b it doesn't appear that C17 heptadecanoic acid was measured in this study, a figure may not be required.

Response: The reviewer's observation regarding the absence of C17 AC detection in the samples is correct. This is due to poor sensitivity of our method assaying C17 AC. The intermediates of long-chain fatty acids with odd number of carbons, including C7, C9, C11, C13, and C15 in Supplemental 2A, represent the metabolites from beta-oxidation of long-chain AC with odd number of carbons. Researchers in the PA field believe that odd-chain fatty acids contribute to propionyl-CoA production. However, it should be noted that one fact about propionyl-CoA production from long-chain fatty acids with odd carbon numbers is overstated. This is because one propionyl-CoA production from these fatty acids is accompanied by more acetyl-CoA synthesis (Supplemental 2B), which offsets the ratio of acetyl-CoA/propionyl-CoA. The authors wish to emphasize this point with this figure. This could be any long-chain fatty acids with odd number of carbons. However, we are open to removing it if the reviewer insists.

11. Propionic acid is reportedly notoriously difficult to measure due to instability, so is your measurement Propionic acid or a derivative 3 hydroxypropionate etc please specify

Response: We greatly appreciate the thorough review provided by this reviewer. We indeed measured the propionate and other short-chain fatty acids. We have included comprehensive methodological details for the analysis of short-chain fatty acids, including propionate, in the revised manuscript's methodology section.

12. Figure 3G is described in the text line 203 stating that propionate emerges the most prominent metabolite exhibiting a substantial decrease this statement should be revised to state that is the most statistically significant based on the P value

Response: We implemented the corrections as per the suggestions provided by this reviewer.

13. Figure 5 is difficult to follow as it doesn't describe M0, the Y axes are similar, figure legend is not very detailed. In the text please comment specifically on which figure that shows 3HIB (line 232). It could be helpful to show a figure with the larger BCAA pathway to orient the reader to the larger picture.

Response: We have revised Figure 5 in accordance with the recommendations made by this reviewer. Additional details explaining M0 have been included in the revised figure legend. More details were added into the BCAA pathway. Reference to the 3HIB labeling in the text can be found in Figure 5D, as indicated in the revised manuscript.

14. Line 241 metabolic substrate of propionyl CoA or propionyl CoA carboxylase?

Response: To enhance clarity, we've revised the phrase "a metabolic substrate of propionyl-CoA" to "a metabolic precursor of propionyl-CoA".

15. Similar comment to Figure 5, difficult to follow. There are several things that are not well described M0, M5. Perhaps some graphs could be moved to supplementals and focus on showing panels that are important/essential to the results and described in the text.

Response: Following the reviewer's suggestion, we have revised the figure accordingly. Certain graphs have been relocated to supplemental Figure 4 for conciseness. Additionally, we have included detailed explanations of stable isotope terminology in figure legends wherever applicable to clarify the stable isotope labeling data. We appreciate your comments in enhancing the clarity of the manuscript.

16. Same comment for Figure 7.

Response: Following the reviewer's suggestion, we have revised the figure accordingly. Certain graphs have been relocated to supplemental Figures 7 and 8 for conciseness. Additionally, we have made similar revisions to the stable isotope terminology in the figure legends to ensure clarity.

17. Figure 7H what was the PCC activity in the mutant mice compared to controls??

Response: It is significantly lower compared to the wild-type control. Please refer to the data below as well as our response to reviewer 2. This difference in comparison has been previously observed in this mouse model. Our primary focus is on investigating the potential regulation of PCC activity by fasting.

Notably, we did not observe any effect of fasting on PCC activity in wild-type mice either (Supplemental Figure 8I), as outlined in our response to reviewer 1, minor comment #1.

18. Line 306 I think you are referring to Suppl Fig 7 not 6 here

Response: Thank you for bringing this to our attention. While it was Supplementary Figure 6 in the initial submission, it now corresponds to Supplementary Figures 8J-8M in the revised manuscript, reflecting the changes made to the supplemental figures during the revision process.

19. Line 328 “correction of metabolic rewiring” is an overstatement

Response: Thanks for this reviewer. We rephrased to “ These metabolic changes during fasting suggested an improvement in the metabolic alterations occurring in PA”.

REVIEWERS' COMMENTS:

Reviewer #2 (Remarks to the Author):

The authors have generally addressed my concerns and I think the manuscript have improved.

However, I still have minor concerns:

Lines 237-238 please reformulate.

Lines 299-303. Please reformulate. I don't think it's a correct statement that there is an increased anaplerosis of the TCA in PA. The low levels of plasma Gln, Ala etc suggest exactly the opposite, that there is a deficient anaplerosis of the TCA. In my opinion, it is highly appreciated the effort of the authors to explain all the metabolic findings, still there are things for which we do not have (yet) an explanation. The complexity of the metabolic pathways in PA, with particularities in every tissue and an important crosstalk between the organs (notably muscle and the liver), reflects the difficulty in the clinical practice to control the metabolic alterations and to find the adequate treatment for our patients.

327 There is no treatment with antibodies in PA to my knowledge.

Reviewer #3 (Remarks to the Author):

Thank you for addressing the majority of my comments. I have re-reviewed the manuscript and it is greatly improved with the edits provided. I commend the authors for addressing comprehensively Major comments #2 and #6, which was also mentioned by the other reviewers, and adding additional limitations to the discussion regarding the applicability of their findings to humans. One additional comment on the revisions please revise the sentence on line 411 "one of the gene mutations observed in human patients with a milder phenotype". Although this statement is true it is misleading given that the references are publications on the mouse model studies not humans. This variant p.Ala138Thr formerly p.Ala113Thr has only been reported in the literature in one individual with propionic acidemia PMID 10780784 and a few entries in ClinVar of unknown affected status. Please consider adding this reference.

RE: Response to Referees (Manuscript COMMSBIO-23-4733A).

REVIEWERS' COMMENTS:

Reviewer #2 (Remarks to the Author):

The authors have generally addressed my concerns and I think the manuscript have improved.

However, I still have minor concerns:

Comment #1: Lines 237-238 please reformulate.

Response: Thank you for highlighting the repetition in the sentence regarding "... with the exception of 3-hydroxyisobutyrate except for 3-hydroxyisobutyrate (3HIB)...". We have revised it by removing the phrase "except for 3-hydroxyisobutyrate".

Comment #2: Lines 299-303. Please reformulate. I don't think it's a correct statement that there is an increased anaplerosis of the TCA in PA. The low levels of plasma Gln, Ala etc suggest exactly the opposite, that there is a deficient anaplerosis of the TCA. In my opinion, it is highly appreciated the effort of the authors to explain all the metabolic findings, still there are things for which we do not have (yet) an explanation. The complexity of the metabolic pathways in PA, with particularities in every tissue and an important crosstalk between the organs (notably muscle and the liver), reflects the difficulty in the clinical practice to control the metabolic alterations and to find the adequate treatment for our patients.

Response: Following the reviewer's recommendation, we have made the necessary revision by eliminating the phrase "increased anaplerosis" from the sentence. We concur with the reviewer's observation regarding the complex nature of metabolic pathways in PA. Omitting "increased anaplerosis" will prevent a definitive interpretation of an intriguing side finding, namely the TCA cycle metabolite labeling data from WAT and brain tissues, as it falls outside the primary focus of this study.

Comment #3: 327 There is no treatment with antibodies in PA to my knowledge.

Response: Thank you for catching that typo. The correct term is "antibiotics" rather than "antibodies". We have made the necessary correction in the revised manuscript.

Reviewer #3 (Remarks to the Author):

Comment #1: Thank you for addressing the majority of my comments. I have re-reviewed the manuscript and it is greatly improved with the edits provided. I commend the authors for addressing comprehensively Major comments #2 and #6, which was also mentioned by the other reviewers, and adding additional limitations to the discussion regarding the applicability of their findings to humans. One additional comment on the revisions please revise the sentence on line 411 "one of the gene mutations observed in human patients with a milder phenotype". Although this statement is true it is misleading given that the references are publications on the mouse model studies not humans. This variant p.Ala138Thr formerly p.Ala113Thr has only been reported in the literature in one individual with propionic acidemia PMID 10780784 and a few entries in ClinVar of unknown affected status. Please consider adding this reference.

Response: Thank you for your feedback. We have incorporated the requested reference into the revised manuscript, now listed as reference #57, as per the reviewer's request.